# DisCoVAE: Disentangling pretrained latent spaces with customized controls using contrastive learning

## Abstract

Deep generative models have recently achieved high-quality results but still lack customizable control abilities. Existing methods mainly rely on using labels as additional inputs to directly condition the generation process. This also constrains the model to be retrained entirely and might ignore the high-level features already captured in the latent space. In this paper, we propose a new approach that allows one to reshape the latent space of pretrained generative models based on user-specified groups of samples. Our method relies on a variational autoencoder trained with an additional supervised contrastive learning regularization. This leads to a new *control* space, which disentangles the features of interest based on the underlying variations found within the custom groups. We propose an iterative approach that can disentangle a single labeled feature at a time from the remaining latent factors without additional supervision, which enables to build the control space gradually. We show that our method outperforms state-of-the-art disentanglement approaches on reference datasets, while also enabling high-quality image synthesis with fine-grained continuous controls on real-world datasets. Our code is available on the supporting webpage [1].

## 1 Introduction

In recent years, deep generative models have demonstrated impressive abilities in modeling complex data distributions, enabling high-fidelity synthesis and powerful representation learning across a wide range of domains (Karras et al., 2019; Ramesh et al., 2022; Borsos et al., 2023). These models mainly rely on *latent*-based approaches such as Variational AutoEncoders (VAE) (Kingma & Welling, 2014), Generative Adversarial Networks (GAN) (Goodfellow et al., 2014), Normalizing Flows (NF) (Rezende & Mohamed, 2015) and Diffusion Models (DM) (Ho et al., 2020), that approximate the underlying data distribution by mapping high-dimensional training data into a lower-dimensional *latent space*. However, these models still lack efficient mechanisms to provide control that aligns with the end-user intent. These latent representations are often difficult to interpret and remain highly entangled which limits their use as straightforward controls. To address this, existing methods usually rely on conditioning techniques that directly provide labeled features as additional inputs during training (Sohn et al., 2015; Lample et al., 2017). However, this ignores the high-level features already captured in the latent space and implies to retrain the model entirely for each new control, which is inefficient as it often requires large amounts of data and computational resources.

In this paper, we propose DisCoVAE (*Disentangling Controls with Variational AutoEncoders*), a new approach that iteratively adds customized controls on top of pretrained generative models without requiring retraining and, therefore, preserving their output quality. We implicitly define controls as the underlying variations found within data groups defined through user-specified labels. We use these groups to train a VAE directly on top of the pretrained latent space to model a new *control space* that disentangles the targeted features.

Pursuing a similar goal, PluGeN (Wołczyk et al., 2022) recently proposed a supervised approach to add controls on pretrained image synthesis models by transforming the entangled latent space into a disen-

---

[1] https://github.com/anonymous/discovae (The GitHub link will be updated upon acceptance notification. The implementation details are already included in the appendix.)

tangled *control-style* space. They split latent features between *control* variables modeled as independent one-dimensional Gaussian mixture distributions explicitly defined by multi-label attributes, and *style* variables supposed to capture the remaining factors of variations. Then, a NF is trained to learn an invertible mapping between both spaces. Although PluGeN performs well on binary attributes, its performance degrades with multi-class attributes and continuous factors, as shown in the original paper. This indicates that the predefined control distributions lack the flexibility to accommodate diverse attribute types.

Instead of using labels to fix the structure of the control space beforehand, we propose to implicitly learn it and rely on user-defined labels to group data samples. This reframes the question of control as the underlying variations found within each of these groups. We instantiate this idea through a separate *variation space*, which is created by computing the element-wise absolute difference between the grouped control embeddings. Hence, this allows us to create multiple views of the original *control space* reflecting each targeted control through different types of variations. We formalize this objective as a contrastive learning problem that aims to cluster variation embeddings representing the same control and separate them from unrelated ones. Applying this contrastive objective to embedding *differences* rather than directly to the embeddings themselves also encourages intra-class separation within each targeted control dimension. Therefore, this leads to modeling the control distributions as independent Gaussian mixtures and transforms discrete attributes into fine-grained continuous controls. This contrastive strategy serves as an additional regularization in the training objective and enables our VAE-based model to directly reshape the pretrained latent space into a new *customized control space* that disentangles the targeted features.

To further personalize the control of pretrained generative models, we generalize our contrastive approach and propose an iterative procedure that enables to disentangle one control at a time using only a single labeled feature. This allows the end-user to efficiently add new controls on a same model while keeping the ones already learned. This also paves the way for extending the approach to real-world datasets that do not necessarily have many label annotations. We show that our iterative approach achieves the highest disentanglement scores and outperforms existing baselines on both reference and real-world image datasets, providing fine-grained control while preserving the high-fidelity synthesis quality of the pretrained model.

Our contributions can be summarized as follows.

- We propose a new framework that combines contrastive learning and generative modeling to enhance pretrained generative models with customized controls.

- We propose a new method that disentangles user-defined features from the pretrained latent factors and transforms discrete labels into fine-grained continuous controls.

- We propose an iterative process that enables one to build the *control space* gradually by disentangling a single labeled feature at a time without additional supervision.

- We provide an invertible mapping between the pretrained latent factors and the user-specified controls which enables to perform *feature extraction*, *editing* and *conditional synthesis* on any generative model with an exposed latent space while maintaining the synthesis quality.

## 2  Related work

Generative models aim to model the underlying distribution $p(\boldsymbol{x})$ of the training data $\boldsymbol{x} \in \mathbb{R}^{d_x}$. Introducing *latent variables* $\boldsymbol{z} \in \mathbb{R}^{d_z}$, defined in a lower-dimensional *latent space* simplifies the complete model to $p(\boldsymbol{x}, \boldsymbol{z}) = p(\boldsymbol{x}|\boldsymbol{z})p(\boldsymbol{z})$. Since this formulation usually has no closed-form solution, VAE uses variational inference to optimize the Evidence Lower Bound (ELBO) as

$$\mathcal{L}_{\text{vae}}(\boldsymbol{\theta}, \boldsymbol{\phi}; \boldsymbol{x}) = \mathbb{E}_{\boldsymbol{z} \sim q_\phi(\boldsymbol{z}|\boldsymbol{x})}\big[\log p_\theta(\boldsymbol{x}|\boldsymbol{z})\big] - \beta \cdot \mathcal{D}_{\text{KL}}\big[q_\phi(\boldsymbol{z}|\boldsymbol{x}) \parallel p(\boldsymbol{z})\big], \tag{1}$$

which balances a reconstruction term and a regularization term. The Kullback-Leibler (KL) divergence encourages the approximate posterior to remain close to a prior distribution $p(\boldsymbol{z})$ by adjusting the $\beta$ hyperparameter. Parametric neural networks are used to model the *encoding* and *decoding* distributions, respectively $q_\phi$ and $p_\theta$.

VAEs have been the primary approach for disentanglement, establishing them as powerful representation learning frameworks (Higgins et al., 2017; Kim & Mnih, 2018; Chen et al., 2018). However, disentangling latent factors often comes at the cost of generative fidelity, highlighting a trade-off between interpretability and visual quality (Burgess et al., 2018). Recently, the authors of TwoStageVAE (Dai & Wipf, 2019) showed that a first VAE can model the underlying data manifold by capturing the necessary latent features to reconstruct the ground truth samples, while a second VAE can be used to model the latent structure by approximating the true probability density measure. This decouples high-fidelity modeling challenges, where the generative models are now mainly based on GANs (Karras et al., 2021) and DMs (Podell et al., 2024), from learning meaningful control representations. Therefore, we aim to rely on this additional VAE strategy to model a disentangled *control space* by leveraging the high-level features extracted by the pretrained generative model. This enables to focus solely on their organization while preserving the synthesis quality of the pretrained model.

The approach most closely related to our work is PluGeN (Wołczyk et al., 2022), which relies on NFs to learn an invertible mapping between the pretrained latent space and a structured *control-style* space where control distributions are pre-defined as independent Gaussian mixtures parameterized by the multi-class labeled attributes. In contrast, we propose to implicitly learn the disentangled control distributions and, therefore, generalize to diverse types of attributes.

Alternatively, recent unsupervised approaches have been proposed to directly explore pretrained latent spaces to uncover interpretable directions that align with the underlying generative factors (Voynov & Babenko, 2020; Ren et al., 2022). In particular, DisCo (Ren et al., 2022) also formalizes a contrastive objective in a *variation space* to discover disentangled control directions. It locally shifts latent points in different orthogonal directions and uses the pretrained generator to generate the associated images. Then, an encoder is trained to uncover the shifted directions that disentangle generative factors by contrasting the embedding distances obtained from the resynthesized images. However, these unsupervised methods assume local linearity and therefore require a certain degree of disentanglement in the pretrained latent space (Karras et al., 2019). Additionally, a post-training analysis is necessary to identify the control directions and a factor can correspond to multiple ones. In contrast, we propose a unified framework that directly combines contrastive learning and generative modeling which generalizes to non-linear latent manifolds while also providing an invertible mapping between the pretrained latent factors and the targeted control features without requiring post-analysis.

## 3 Proposed method

We present our VAE-based model that iteratively disentangles user controls by reshaping pretrained latent spaces with contrastive learning. Fig. 1 illustrates the overall approach.

### 3.1 Problem statement - Defining the control space

We consider a dataset of training examples $\boldsymbol{x} \in \mathbb{R}^{D_x}$ paired with $K$ discrete multi-labels $\boldsymbol{y} = (y_1, ..., y_K) \in [\![1, M_k]\!]^K$, where $M_k$ is the number of classes for each attribute $y_k$. These attributes define a subset of the $D$ underlying generative factors of $\boldsymbol{x}$ ($K \leq D$). Our goal is to model a disentangled *control space* such that the continuous control vector $\boldsymbol{c}_{\leq K} = (c_1, ..., c_K)$ can be defined by the factorial distribution of $K$ independent continuous control variables modeling each discrete attribute

$$p(\boldsymbol{c}_{\leq K}) = \prod_{k=1}^{K} p(c_k), \tag{2}$$

where $p(c_k) = \sum_{j=1}^{M_k} p(c_k \mid y_{kj}) \, p(y_{kj})$.

To achieve this, we train a VAE on top of the pretrained latent space $\boldsymbol{z} \in \mathbb{R}^{D_z}$ ($K \leq D \leq D_z$), which preserves the latent dimensionality to retain the high-level features extracted by the pretrained generative model and focus solely on their organization. Thus, we further refine our objective from Eq. 2 by introducing the variables $\boldsymbol{c}_{>K} = (c_{K+1}, ..., c_{D_z})$ that aim to capture the remaining factors of variations. Hence, the *control*

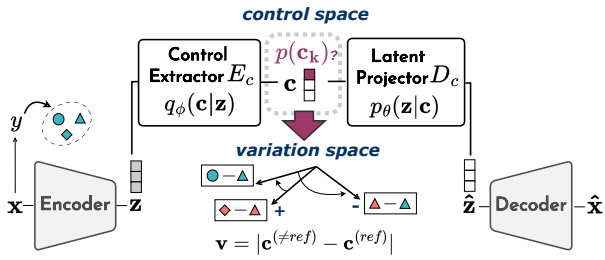

(a) **Method 1**: Creating the *variation space* by computing the element-wise absolute difference between the grouped control embeddings to implicitly model $p(c_k)$.

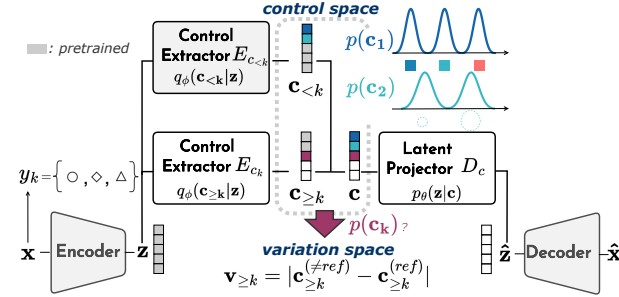

(b) **Method 2**: Disentangling a single labeled feature to iteratively build the *control space* on a same model.

Figure 1: Detailed overview of our approach that disentangles user-defined controls from pretrained latent factors by contrasting the grouped embeddings in the *variation space* to model the control distributions $p(c_k)$ as independent Gaussian mixtures.

*space* is defined by the entire vector $\boldsymbol{c} = (c_1, ..., c_K, c_{K+1}, ..., c_{D_z}) = (\boldsymbol{c}_{\leq K}, \boldsymbol{c}_{>K}) \in \mathbb{R}^{D_z}$ and our objective is to model the disentangled prior distribution

$$p(\boldsymbol{c}) = \prod_{k=1}^{K} p(c_k) \cdot p(\boldsymbol{c}_{>K}). \tag{3}$$

As depicted in Fig. 1a, the latent encoder $E_{\boldsymbol{c}}$, denoted as *control extractor*, models $q_\phi(\boldsymbol{c}|\boldsymbol{z})$, while the decoder $D_{\boldsymbol{c}}$, referred to as *latent projector*, maps the control representation back to the initial latent space by approximating $p_\theta(\boldsymbol{z}|\boldsymbol{c})$. Hence, we can rely on the ELBO in Eq. 1 and optimize

$$\mathcal{L}(\boldsymbol{\theta}, \boldsymbol{\phi}; \boldsymbol{z}) = \mathbb{E}_{\boldsymbol{c} \sim q_\phi(\boldsymbol{c}|\boldsymbol{z})} \big[ \log p_\theta(\boldsymbol{z} \mid \boldsymbol{c}) \big] - \beta \cdot \mathcal{D}_{\mathrm{KL}} \big[ q_\phi(\boldsymbol{c} \mid \boldsymbol{z}) \parallel p(\boldsymbol{c}) \big]. \tag{4}$$

We extend this objective with a contrastive learning regularization to model the unknown prior $p(\boldsymbol{c})$.

### 3.2 Disentangling controls via contrastive learning

The core idea of our approach lies in defining controls implicitly as the underlying variations within groups of examples. Our method explicitly assigns these variations to embedding dimensions by contrasting different views of each control feature in a *variation space* that captures relative differences between the grouped control embeddings.

For illustrative purposes, consider a simple dataset composed of circles, squares, and triangles with 2 different sizes and 3 colors. The *shape* control could be defined as the varying feature inside groups of examples with same size and color but different shapes, while the *color* control would be represented through groups with same shape and size but different colors. If we select a unique example in each group and compute the embedding absolute difference with respect to this *reference*, we can create different views of the underlying variations for a same feature. We can then apply a contrastive objective on this *variation space* to encourage the model to bring closer the views associated with the same control, considered as *positive* pairs, while pushing apart those from different controls, denoted as *negative* pairs.

In VAEs, the mean-field variational family assumes mutually independent latent variables designed through the encoder predicting the parameters of a multivariate Gaussian distribution with a diagonal covariance matrix. Hence, extending the VAE framework with this discriminative objective forces the model to encode controls into independent dimensions. This enables modeling a disentangled control space and provides an alternative to the manually-designed orthogonal projection matrix in DisCo (Ren et al., 2022).

### 3.2.1 Method 1 (M1): disentangling multiple controls at once.

Formally, we gather the training data $\boldsymbol{x}$ into groups $\mathcal{G}_k$ that provide individual views of each control variable $c_k$. Each group corresponds to the set of examples $\boldsymbol{x}$ whose associated multi-labels $\boldsymbol{y} = (y_1, ..., y_K)$ are all identical except for the $y_k$ component which varies. For each $c_k$, we denote as $\mathcal{X}_k$ the set of all these groups.

In each training step, we consider a batch of $N$ groups. First, we focus on each control representation individually. From each group $\mathcal{G}_k \in \mathcal{X}_k$, we randomly sample an example $\boldsymbol{x}_k^{(\mathrm{ref})}$ to serve as *reference* anchor. We use the pretrained encoder $E_{\boldsymbol{z}}$ to sample the corresponding latent representations $\boldsymbol{z}_k \in \mathbb{R}^{D_z}$ such that $\boldsymbol{z}_k \sim \mathcal{N}(\mu_{\mathbf{z}_k}, \boldsymbol{\Sigma}_{\mathbf{z}_k})$, where $(\mu_{\mathbf{z}_k}, \boldsymbol{\Sigma}_{\mathbf{z}_k}) = E_{\boldsymbol{z}}(\boldsymbol{x}_k)$ and $\boldsymbol{x}_k \in \mathcal{G}_k$ (also including $\boldsymbol{x}_k^{(\mathrm{ref})}$). The latent vectors are then fed to the *control extractor* $E_{\boldsymbol{c}}$ which outputs the parameters of the multivariate Gaussian density $q_\phi$, $(\boldsymbol{\mu}_{\boldsymbol{c}_k}, \boldsymbol{\Sigma}_{\boldsymbol{c}_k}) = E_{\boldsymbol{c}}(\boldsymbol{z}_k)$, from which we sample the control embeddings

$$\mathbf{c}_k \sim \mathcal{N}(\boldsymbol{\mu}_{\boldsymbol{c}_k}, \boldsymbol{\Sigma}_{\boldsymbol{c}_k}). \tag{5}$$

Then, we compute the associated variation embeddings $\boldsymbol{v}_k \in \mathbb{R}^{D_z}$ at the groups level, defined as

$$\boldsymbol{v}_k = |\boldsymbol{c}_k^{(\neq \mathrm{ref})} - \boldsymbol{c}_k^{(\mathrm{ref})}|, \tag{6}$$

which corresponds to the element-wise absolute difference between the control embedding of the reference $\boldsymbol{x}_k^{(\mathrm{ref})}$ and the control embeddings of the remaining examples $\boldsymbol{x}_k \in \mathcal{G}_k \setminus \{\boldsymbol{x}_k^{(\mathrm{ref})}\}$. Hence, the set of variation embeddings $\boldsymbol{v}_k$ from all $\mathcal{G}_k \in \mathcal{X}_k$ defines the *variation space* $\mathcal{V}_k$ that represents different views of $c_k$.

From this, we formalize our contrastive learning objective and build the *query*, *positive* and *negative* sets for each control. Each variation set $\mathcal{V}_k$ is randomly split into a query set $\mathcal{Q}_k$ and a positive set $\mathcal{K}_k^+$, of equal sizes, to construct query-positive *variation* pairs that model the control variable $c_k$. To build the negative set $\mathcal{K}_k^-$, we sample variation embeddings from the other variation sets $\mathcal{V}_l$ ($l \neq k$) where the target control feature remains constant (identical $y_k$). To enforce each control to be exactly encoded in a single dimension, we propose to create $\mathcal{K}_k^-$ directly from $\mathcal{V}_k$ and replace only the $K$ first dimensions with embedding values from other $\mathcal{V}_l$. Finally, we can formalize the InfoNCE loss proposed in (Oord et al., 2018; Chen et al., 2020) as

$$\mathcal{L}_{\mathrm{InfoNCE}}(\boldsymbol{\phi}; \boldsymbol{v}) = -\frac{1}{K} \sum_{k=1}^{K} \frac{1}{|\mathcal{Q}_k|} \sum_{(\boldsymbol{v}_q, \boldsymbol{v}_p) \in (\mathcal{Q}_k, \mathcal{K}_k^+)} \log \frac{\exp(\mathrm{sim}(\boldsymbol{v}_q, \boldsymbol{v}_p)/\tau)}{\exp(\mathrm{sim}(\boldsymbol{v}_q, \boldsymbol{v}_p)/\tau) + \sum_{\boldsymbol{v}_n \in \mathcal{K}_k^-} \exp(\mathrm{sim}(\boldsymbol{v}_q, \boldsymbol{v}_n)/\tau)}, \tag{7}$$

where $\boldsymbol{v}_q$, $\boldsymbol{v}_p$, and $\boldsymbol{v}_n$ are, respectively, the query, positive, and negative variation embeddings, $\mathrm{sim}(\cdot, \cdot)$ corresponds to the cosine similarity, and $\tau$ is the temperature hyperparameter between 0 and 1 that modulates the softmax distribution by scaling the similarity scores.

Although this promotes features disentanglement, it does not ensure a continuous prior $p(\boldsymbol{c})$ from which we can sample to generate new examples because the contrastive objective is unbounded. To address this, we continue to optimize the KL divergence between the approximate posterior $q_\phi$ and a standard isotropic normal $\mathcal{N}(\mathbf{0}, \boldsymbol{I})$. This restrains the ranges of the control space, and therefore models $p(\boldsymbol{c}_{\leq K})$ as a set of independent Gaussian mixtures thanks to our contrastive strategy that implicitly promotes intra-class separation. This also enables us to keep the remaining $D_z - K$ latent factors in the control space, required to reconstruct the input latent vectors.

Hence, our method successfully combines both generative and discriminative frameworks from Eq. 4 and Eq. 7 to model a continuous disentangled control space. The complete training objective is defined as

$$\mathcal{L}(\boldsymbol{\theta}, \boldsymbol{\phi}; \boldsymbol{z}) = -\mathbb{E}_{\boldsymbol{c} \sim q_\phi(\boldsymbol{c}|\boldsymbol{z})} \big[ \log p_\theta(\boldsymbol{z} \mid \boldsymbol{c}) \big] + \beta \cdot \mathcal{D}_{\mathrm{KL}} \big[ q_\phi(\boldsymbol{c} \mid \boldsymbol{z}) \parallel \mathcal{N}(\mathbf{0}, \boldsymbol{I}) \big] + \mathcal{L}_{\mathrm{InfoNCE}}(\boldsymbol{\phi}; \boldsymbol{v}). \tag{8}$$

However, this approach requires knowing all the generative factors beforehand. Moreover, it only holds for $K \geq 2$ because the negative examples are sampled from the embeddings variation sets representing at least another control which ensures that the targeted control dimension remains constant. Additionally, relying solely on existing combinations of attributes may introduce class imbalance and biases compromising generalization and disentanglement quality. To address this, we generalize this method to the case where $K = 1$, which enables us to disentangle one control at a time using a single labeled feature without knowing the other generative factors.

### 3.2.2 Method 2 (M2): disentangling one control at a time.

We now consider a single control variable $c_k$ and propose a strategy to build the *variation space* $\mathcal{V}_k = \mathcal{Q}_k \cup \mathcal{K}_k^+$ directly from a batch of examples while keeping its properties for disentanglement. We create multiple views of $c_k$ by contrasting *cross-class* embeddings with *same-class* ones, and design symmetric negative pairs to cancel out the other factors across dimensions. We illustrate our approach on a simple case in Fig. 2.

Formally, we consider a batch of N examples $\mathcal{X} = \{\boldsymbol{x}_i\}_{i=1}^N$ paired with discrete labels $\mathcal{Y} = \{y_i\}_{i=1}^N$, with $y_i \in [\![1, M_k]\!]$. We denote $\mathcal{A} \subseteq [\![1, M_k]\!]$ the subset of class labels available in the current batch. In each iteration, we sample the control embeddings $\boldsymbol{c}_i$ for each example $\boldsymbol{x}_i$ as defined in Eq. 5. Then, we rely on the assigned labels to randomly partition the control embeddings into two disjoint subsets, $\mathcal{C}^Q$ and $\mathcal{C}^+$, so that each class is equally represented (i.e., each subset contains the same number of examples from every class). For each pair of embeddings

$$(\boldsymbol{c}_{i_1}, \boldsymbol{c}_{i_2}) \in \mathcal{C}^Q \times \mathcal{C}^+ \quad \text{such that} \quad y_{i_1} = y_{i_2} = i \in \mathcal{A},$$

where $i$ corresponds to the *reference* class (e.g. *triangle* in Fig. 2), we randomly assign another pair of embeddings

$$(\boldsymbol{c}_{j_1}, \boldsymbol{c}_{j_2}) \in \mathcal{C}^Q \times \mathcal{C}^+ \quad \text{such that} \quad y_{j_1} \neq i \text{ and } y_{j_2} \neq i.$$

No additional constraint is imposed on $y_{j_1}$ and $y_{j_2}$, i.e., they may be equal or different. The only requirement is that neither belongs to the reference class $i$. We then compute the query $\boldsymbol{v}_q \in \mathcal{Q}_k$ and positive $\boldsymbol{v}_p \in \mathcal{K}_k^+$ variation embeddings, respectively, as

Figure 2: (Method 2) Modeling the *shape* control distribution $p(c_1)$ in the first embedding dimension without knowing the other factors (e.g. *color* and *size*). The query $\mathcal{Q}_1$ and positive $\mathcal{K}_1^+$ sets capture similar variations of shapes (circles or squares against triangles selected as *reference* embeddings). The negative set $\mathcal{K}_1^-$ inherits from $\mathcal{Q}_1$ and $\mathcal{K}_1^+$ and removes *shape* variations from the first embedding dimension.

$$\boldsymbol{v}_q = |\boldsymbol{c}_{i_1} - \boldsymbol{c}_{j_1}| \quad \text{and} \quad \boldsymbol{v}_p = |\boldsymbol{c}_{i_2} - \boldsymbol{c}_{j_2}|. \tag{9}$$

Therefore, the targeted feature effectively varies in both the query $\mathcal{Q}_k$ and positive $\mathcal{K}_k^+$ variation sets in order to represent different views of the control.

To ensure that the control variable $c_k$ captures only variations related to the targeted labeled feature in dimension $k$, we manually construct the negative variation embeddings $\boldsymbol{v}_n \in \mathcal{K}_k^-$ from the same control embeddings such that, $\forall d \in \{1, \dots, D_z\}$,

$$\boldsymbol{v}_{n_1} = |\boldsymbol{c}_{i_1} - \boldsymbol{c}_p|, \quad \text{with} \qquad \boldsymbol{c}_p = \begin{cases} \boldsymbol{c}_{i_2}, & \text{if } d = k \\ \boldsymbol{c}_{j_2}, & \text{if } d \neq k \end{cases}, \tag{10}$$

and similarly,

$$\boldsymbol{v}_{n_2} = |\boldsymbol{c}_{i_2} - \boldsymbol{c}_q|, \quad \text{with} \qquad \boldsymbol{c}_q = \begin{cases} \boldsymbol{c}_{i_1}, & \text{if } d = k \\ \boldsymbol{c}_{j_1}, & \text{if } d \neq k \end{cases}. \tag{11}$$

Each *query-positive* pair $(\boldsymbol{v}_q, \boldsymbol{v}_p)$ yields two *query-negative* pairs, $(\boldsymbol{v}_q, \boldsymbol{v}_{n_1})$ and $(\boldsymbol{v}_q, \boldsymbol{v}_{n_2})$. First, we remove variations of the target control from the negative variation embeddings by comparing *same-class* control embeddings along dimension $k$. Second, the symmetric design of the remaining embedding components ensures that variations unrelated to the target control do not influence the minimization of the InfoNCE loss as defined in Eq. 7. We formalize this property in Theorem 1. Hence, the contrastive objective ignores any representation component that does not help discriminate positives from negatives. Therefore, the contrastive regularization exclusively focuses on disentangling the labeled feature from the remaining latent factors and ensures that variations of the target control are only encoded in dimension $k$.

Finally, we train our model with the same training objective defined in Eq. 8 with $K = 1$. The algorithms are detailed in the appendix A.1.

**Theorem 1** (The symmetric design of embeddings isolates the optimization of the target control dimension)**.** *Let $\boldsymbol{v}_q$, $\boldsymbol{v}_p$, $\boldsymbol{v}_{n_1}$, and $\boldsymbol{v}_{n_2}$ be the variation embeddings defined in Eqs. 9–10–11. Assume that $(\boldsymbol{c}_{j_1}, \boldsymbol{c}_{j_2})$ are sampled independently subject only to the constraint $y_{j_1} \neq i$ and $y_{j_2} \neq i$.*

*Then for any dimension $d \neq k$, the expected gradient of the InfoNCE loss with respect to the corresponding control variable vanishes such that*

$$\mathbb{E}_{(\boldsymbol{c}_{j_1}, \boldsymbol{c}_{j_2})}\left[\nabla_{c_d}\mathcal{L}_{InfoNCE}\right] = 0.$$

*Consequently, in expectation, the optimization of the contrastive objective is not influenced by variations along non-target dimensions.*

*Proof.* Consider a dimension $d \neq k$. By construction, the components used to build the variation embeddings along dimension $d$ are obtained from $\boldsymbol{c}_{i_1}, \boldsymbol{c}_{i_2}, \boldsymbol{c}_{j_1}$, and $\boldsymbol{c}_{j_2}$.

Since $(\boldsymbol{c}_{j_1}, \boldsymbol{c}_{j_2})$ are sampled without additional constraints on their class labels (except $y_{j_1} \neq i$ and $y_{j_2} \neq i$), the corresponding components are drawn symmetrically from the same distribution in the positive and negative constructions.

Therefore the distributions of the components $v_{p,d}$, $v_{n_1,d}$, and $v_{n_2,d}$ are identical, which implies

$$\mathbb{E}[v_{p,d}] = \mathbb{E}[v_{n_1,d}] = \mathbb{E}[v_{n_2,d}].$$

The InfoNCE loss depends on these quantities only through similarity comparisons between positive and negative pairs. Identical expected contributions therefore imply that the expected derivative of the loss with respect to $c_d$ cancels out, yielding

$$\mathbb{E}_{(\boldsymbol{c}_{j_1}, \boldsymbol{c}_{j_2})}\left[\nabla_{c_d}\mathcal{L}_{\text{InfoNCE}}\right] = 0 \quad \text{for } d \neq k.$$

$\square$

This result implies that, in expectation, the optimization signal of the InfoNCE objective is concentrated on the target dimension $k$, while the remaining dimensions contribute only symmetrically and therefore do not bias the learning dynamics.

### 3.3 Building the control space iteratively

The ability to disentangle a single labeled control feature from pretrained latent spaces without additional supervision offers two main advantages. First, we can address specific control challenges by adapting the training configuration (e.g. batch size, number of training steps, data balancing), and therefore, support diverse control types. Second, this allows us to formalize an incremental procedure to iteratively add new controls on a same model as depicted in Fig 1b. This enables the end-user to build the control space gradually from different sets of custom labels allowing further personalization of pretrained generative models. This is made efficient by the use of a lightweight VAE architecture that can be easily retrained. As illustrated in Fig 1b, the pretrained encoder $E_{\mathbf{c}_{<k}}$ infers the $(k-1)$-first control dimensions $\mathbf{c}_{<k}$ that are already disentangled. We initialize the control extractor $E_{\mathbf{c}_{\geq k}}$ associated to the new feature using the pretrained weights. This encoder models $q_\phi(\mathbf{c}_{\geq k}|\mathbf{z})$ and only focuses on disentangling the new target control variable $c_k$ optimizing the contrastive objective on the $D_z - k$ remaining dimensions while the same latent projector tries to reconstruct the input latent vectors from the mixed control embeddings concatenating both pretrained and currently-optimized dimensions.

## 4 Experiments

We aim to assess the abilities of our method to (1) accurately disentangle the selected features from the pretrained latent space and (2) effectively provide fine-grained continuous controls to perform *feature extraction*, *editing* and *conditional synthesis* on pretrained generative models without degrading the pretrained output quality.

**Datasets** We evaluate on two standard disentanglement datasets with both categorical and continuous generative factors. **DSprites** (Matthey et al., 2017) consists of 737280 grayscale images generated from five ground-truth factors: *shape* (3 classes), *scale* (6), *orientation* (40), *x-position* (32), and *y-position* (32). **3DShapes** (Burgess & Kim, 2018) includes 480000 images varying across six factors: *floor color*, *wall color*, *object color* (10 each), *scale* (8), *shape* (4), and *orientation* (15). For M1, we preprocess groups of samples where only one factor varies. For real-world evaluation, we use **Flickr-Faces-HQ** (**FFHQ**) (Karras et al., 2019), a dataset of high-resolution images. We select the same 10000 images as PluGeN annotated via the Microsoft Face API (Abdal et al., 2021) with 8 attributes: *gender*, *eyeglasses*, *pitch*, *yaw*, *baldness*, *facial hair*, *age*, and *expression*. Continuous attributes are quantized into discrete classes, yielding 2, 2, 6, 6, 2, 2, 8, and 2 classes, respectively.

**Evaluation metrics** We follow the well-accepted protocol (Locatello et al., 2019) to evaluate disentanglement on reference datasets and consider two standard metrics. The *Disentanglement-Completeness-Informativeness* (DCI) score (Eastwood & Williams, 2018) measures how well each latent dimension captures a single factor, and the *Mutual Information Gap* (MIG) (Chen et al., 2018) measures the gap in mutual information between the top two latent dimensions associated with each ground-truth factor. MIG indicates whether a factor is primarily captured by a single latent dimension. Both metrics, between 0 and 1, are complementary and higher scores indicate better disentanglement. All reported scores are averaged over 5 random seeds. To assess synthesis quality, we compute the *Frechet Inception Distance* (FID) (Heusel et al., 2017) using the standard `clean-fid` library (Parmar et al., 2022). Lower FID means higher fidelity and more realistic outputs. We evaluate our method's ability to control image generation by accurately extracting features and evaluating the consistency of re-synthesized images under feature modifications. For *feature extraction*, we compute a prediction accuracy score by assigning inferred control variables to the nearest Gaussian mean of each mixture $p(c_k)$ and comparing it with the associated ground-truth labels. For *attribute editing*, we extract the control representations from input images, randomly swap attributes (varying the number of changed attributes), and measure prediction accuracy using a pretrained classifier (with CLIP (Radford et al., 2021) embeddings for FFHQ). For *conditional synthesis*, we sample random combinations of the mixture and compute control accuracy on the synthesized images with these classifiers.

**Implementation details** We pre-trained convolutional VAEs on DSprites and 3DShapes for 1M and 300k steps, respectively, using $\beta = 1$ with 10 and 64 latent dimensions and architectures similar to DisCo's encoders. For FFHQ, we used the official DiffAE model (Preechakul et al., 2022) pretrained on 256x256 images with 512 latent dimensions. We train our lightweight VAE-based models on these pretrained latent spaces using Adam optimizer with a constant learning rate of $10^{-4}$ and same architectures as in TwoStageVAE resulting in 4.2M, 4.5M and 7.6M trainable parameters for each dataset respectively. (See Appendix.) For DSprites, models are trained for 300k steps with $\beta = 0.05$ and batch size of 32 groups for M1 and 512 for M2. For 3DShapes, models are trained for 300k steps with $\beta = 0.001$ and batch size of 64 groups for M1 and $\beta = 0.01$ and batch size of 256 for M2. For FFHQ, models are trained for 40k steps for each control with $\beta = 0.0005$ and batch size of 1024. The temperature $\tau$ is set to 0.07 as in (Chen et al., 2020). Since the dimensionality of the pretrained latent space of DiffAE is relatively high (512 dimensions), we observed in practice that slightly relaxing the KL regularization on the target control dimension helps guide the model to disentangle the targeted feature and facilitates convergence when multiple controls are added iteratively ($k \geq 2$ in M2). This effect was particularly noticeable for attributes whose classes are highly imbalanced in the dataset, such as *glasses* or *baldness*. Formally, we can rewrite the KL regularization term in eq. 8 as $\beta_k \cdot \mathcal{D}_{\mathrm{KL}}(q_\phi(c_k \mid z) \,\|\, \mathcal{N}(0,1)) + \beta_{\neq k} \cdot \mathcal{D}_{\mathrm{KL}}(q_\phi(\mathbf{c}_{\neq k} \mid z) \,\|\, \mathcal{N}(\mathbf{0}, \mathbf{I}))$, where slightly different regularization strengths are applied for the target and non-target dimensions with $\beta_k \leq \beta_{\neq k}$. However, the gap between the two coefficients should remain small enough, so that the variational mean-field assumption of independent latent dimensions remains valid. For instance, we set $\beta_{\neq k} = 0.0005$ and $\beta_k = 0.0001$.

**Baselines** For disentanglement, we compare our methods to *TwoStageVAE* which can also be considered as an ablation study removing the contrastive regularization from the ELBO, and *DisCo* considering all latent dimensions or only the exact number of ground-truth factors as candidate directions for the navigator. For control, we compare to *PluGeN*, which also enables feature extraction, editing and conditional synthesis. We train for 300k steps using the official implementations on the same pretrained generative models.

Table 1: Disentanglement results on **DSprites** ($a = 10$, $b = 5$) and **3DShapes** ($a = 64$, $b = 6$) datasets.

| | Disentanglement | | Quality | Disentanglement | | Quality |
|---|---|---|---|---|---|---|
| | DCI ↑ | MIG ↑ | FID ↓ | DCI ↑ | MIG ↑ | FID ↓ |
| dataset | DSprites | | | 3DShapes | | |
| pretrained | $0.131 \pm 0.001$ | $0.043 \pm 0.002$ | **49.928** | $0.581 \pm 0.003$ | $0.222 \pm 0.001$ | 5.338 |
| TwoStageVAE | $0.083 \pm 0.002$ | $0.015 \pm 0.001$ | 60.513 | $0.671 \pm 0.001$ | $0.139 \pm 0.001$ | 28.254 |
| DisCo ($a$ dir.) | $0.069 \pm 0.001$ | $0.024 \pm 0.003$ | - | $0.810 \pm 0.003$ | $0.130 \pm 0.004$ | - |
| DisCo ($b$ dir.) | $0.051 \pm 0.002$ | $0.022 \pm 0.001$ | - | $0.583 \pm 0.004$ | $0.208 \pm 0.001$ | - |
| **Ours** (M1 $a$) | $0.683 \pm 0.001$ | $0.491 \pm 0.002$ | 50.799 | **0.999** $\pm 0.000$ | $0.174 \pm 0.001$ | 5.287 |
| **Ours** (M1 $b$) | $0.550 \pm 0.002$ | $0.399 \pm 0.008$ | 50.845 | $0.490 \pm 0.006$ | $0.010 \pm 0.001$ | 5.363 |
| **Ours** (M2) | **0.755** $\pm 0.003$ | **0.654** $\pm 0.002$ | 51.557 | **0.999** $\pm 0.000$ | **0.816** $\pm 0.001$ | **5.264** |

# 5 Results

## 5.1 Control space disentanglement

First, we evaluate our method's ability to disentangle control features from the pretrained latent factors. We report the results in Table 1 for both DSprites and 3DShapes. On both datasets, our method strongly outperforms all baselines and achieves high disentanglement scores without degrading the pretrained synthesis quality as highlighted by FID scores that remain comparable to the VAE reference and even improve on 3DShapes. This aligns with theoretical observations made in (Dai & Wipf, 2019) that VAEs can achieve high-fidelity synthesis like GANs if the latent space successfully captures the true probability density measure of the underlying data manifold, which is the case when all of the generative factors are disentangled.

As shown in Table 1 on DSprites, when the initial pretrained latent space is fully entangled, baseline methods fail to recover meaningful factors of variation. This is especially the case for DisCo, which performs orthogonal linear shifts around latent points to uncover directions of variations and, therefore, implicitly assumes local linearity. In contrast, our approach is more robust and improves disentanglement even in highly nonlinear latent manifolds. This highlights the benefits of the contrastive learning regularization in the ELBO as compared to TwoStageVAE. The proportion of latent dimensions to control factors also significantly influences performance for both DisCo and our method M1 with the preprocessed groups. When imposing the number of control dimensions to be equal to the exact number of generative factors, disentanglement scores degrade and it even leads to posterior collapse for our method when trained on 6 dimensions for 3DShapes. While our unconstrained M1 approach achieves the highest DCI score on 3DShapes, MIG remains relatively low indicating that features are still encoded into multiple dimensions, which precludes straightforward controls and necessitates a post-training analysis. Conversely, our method M2 outperforms all models in both DCI and MIG scores indicating that staged training improves stability and generalization. This supports our iterative proposed approach and confirms that learning one control at a time can be achieved using a single labeled feature without prior knowledge of the remaining generative factors, while also yielding better disentanglement performance.

## 5.2 Controlling pretrained generative models

Finally, we evaluate the control abilities of our iterative approach and show that our method outperforms the baselines on 1) *feature extraction*, 2) *conditional synthesis*, and 3) *attribute editing* tasks.

### 5.2.1 Feature extraction

As highlighted in Table 2 and Table 3, our method achieves significantly better feature extraction accuracy scores than PluGeN on reference datasets and comparable results on FFHQ. This empirically shows that enforcing a predefined disentangled prior does not necessarily lead to disentangled representations in practice, whereas our contrastive objective successfully enforces intra-class separation. PluGeN restrains the Gaussian mixtures' means between $-1$ and $1$. Although this might be suited for binary attributes, which are used

Table 2: Feature extraction accuracy (%) on DSprites (top) and 3DShapes (bottom).

| Attr. | Classif. | PluGeN | Ours |
|-------|----------|--------|------|
| Shape | **100.00** | 79.04 | 99.99 |
| Scale | **99.67** | 68.89 | 99.23 |
| Orientation | **86.29** | 2.93 | 4.46 |
| Position X | **98.99** | 15.03 | 81.55 |
| Position Y | **98.75** | 13.89 | 83.61 |
| **Mean** | **96.74** | 35.96 | 73.77 |
| Floor Color | **100.00** | 35.39 | 99.94 |
| Wall Color | 100.00 | 61.43 | **100.00** |
| Object Color | **100.00** | 32.54 | 99.98 |
| Object Size | 99.95 | 65.50 | **99.98** |
| Object Type | 100.00 | 91.43 | **100.00** |
| Orientation | **100.00** | 31.50 | 99.22 |
| **Mean** | **99.99** | 52.97 | **99.85** |

Table 3: Feature extraction accuracy (%) for each attribute on FFHQ. We compare our pretrained multi-head classifier (Classif.) and the PluGeN baseline against our M2 models trained for a single-control ($c1$) and our M2 multi-control model ($c_K$) built iteratively.

| Attr. | Classif. | PluGeN | **Ours** ($c_1$) | **Ours** ($c_K$) |
|-------|----------|--------|------------------|------------------|
| Gender | **96.15** | 93.11 | 95.68 | 95.68 |
| Glasses | 98.95 | 98.81 | 98.86 | **98.96** |
| Yaw | 43.50 | 75.15 | 71.83 | **75.94** |
| Pitch | 44.05 | **55.26** | 54.17 | 53.17 |
| Bald. | **98.75** | 98.41 | 97.97 | 97.57 |
| Fac.Hair | **97.45** | 96.88 | 96.33 | 96.97 |
| Age | **66.75** | 59.23 | 59.42 | 61.56 |
| Expr. | 94.40 | 94.10 | 95.04 | **95.09** |
| **Mean** | 79.50 | 83.87 | 83.66 | **84.37** |

on FFHQ and might explain the high prediction score, accuracy decreases with more classes as observed for 3DShapes (up to 15 classes) and DSprites (up to 40 classes). In contrast, our method is comparatively more robust to variations across attribute types, except for the *orientation* factor in DSprites. Notably, our method achieves performance comparable to the multi-head classifier and even outperforms it on FFHQ, suggesting strong disentanglement and effective class separation for each control.

### 5.2.2 Conditional synthesis

As shown in Table 4, the same observation holds for conditional synthesis where controls are directly sampled from the control space. While PluGeN yields better results on direct reconstruction task as normalizing flows provide exact invertible mapping, its performance severely degrades on randomly synthesized images. In contrast, our method remains robust to conditional sampling (see appendix A.3.1). This shows that our contrastive objective encourages the control extractor to learn informative dimensions, which also prevents posterior collapse and improves latent reconstruction. Hence, our method alleviates architectural constraints, ensures more stable training, and opens the possibility for extending our approach to other data modalities.

### 5.2.3 Attributes Editing

As depicted in Fig. 3, our method exhibits greater resilience than PluGeN to attributes swapping in terms of both control accuracy and FID scores on re-synthesized images. In particular, we maintain near-perfect accuracy on 3DShapes even when all attributes are swapped while PluGeN's scores already drop when swapping only two attributes. This shows that our method can provide strong independence between attributes in the control space and aligns with the high disentanglement scores presented in Table 1. This also provides evidence in favor of learning control distributions implicitly from pretrained latent factors whereas pre-defining the ordering of Gaussians in the mixture may be challenging for multi-class categorical labels. While our method demonstrates similar control accuracy on FFHQ, the performance gap in FID scores is substantial showing that PluGeN fails to maintain the synthesis quality of the pretrained model, also

Table 4: Control accuracy (%) on generated images for direct reconstruction and conditional synthesis tasks.

| | DSprites | | | | 3DShapes | | | | FFHQ | | | |
|---|---|---|---|---|---|---|---|---|---|---|---|---|
| | Rec. | | C. Synth. | | Rec. | | C. Synth. | | Rec. | | C. Synth. | |
| | Acc.↑ | FID↓ | Acc.↑ | FID↓ | Acc.↑ | FID↓ | Acc.↑ | FID↓ | Acc.↑ | FID↓ | Acc.↑ | FID↓ |
| PluGeN | **92.11** | **49.9** | 20.41 | 60.7 | **99.92** | 5.3 | 36.54 | 16.6 | **77.65** | **9.9** | 49.54 | 409.2 |
| **Ours** | 87.15 | 51.6 | **67.47** | **58.5** | 99.91 | 5.3 | **82.64** | **12.1** | 75.88 | 26.5 | **74.04** | **85.7** |

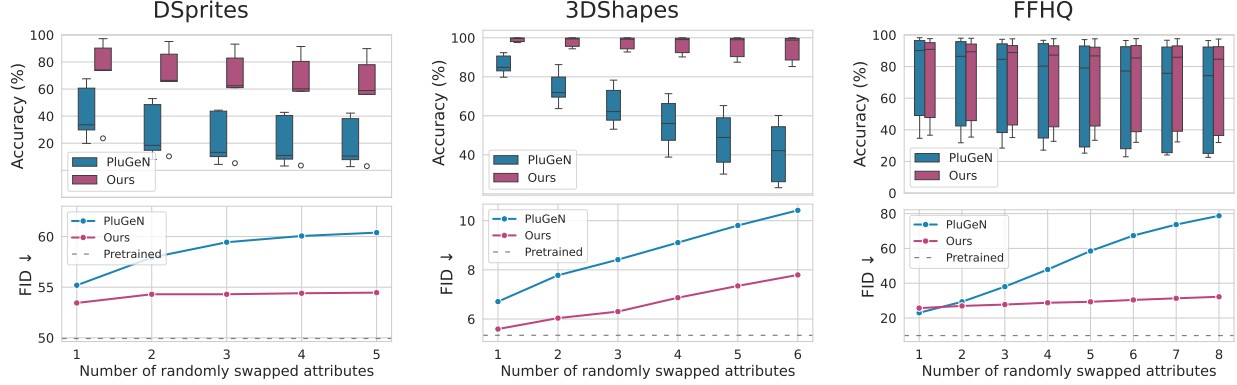

Figure 3: Comparison of our method with PluGeN for attributes editing. Evaluating control prediction accuracy and FID score on resynthesized images while increasingly randomly swapping one to all features.

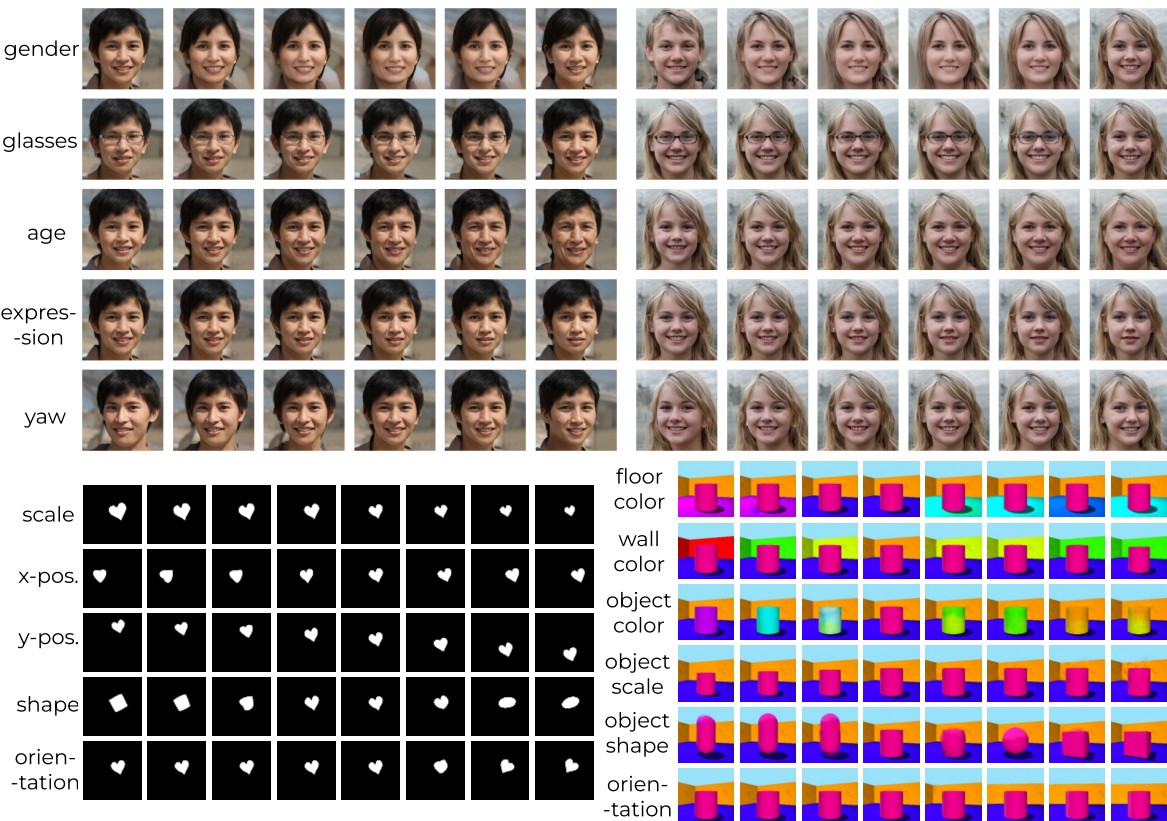

Figure 4: Traversals in the control space of our iterative method M2 on FFHQ, DSprites and 3DShapes.

observed in Table 4. In contrast, FID remains stable for our method, which successfully preserves the high-fidelity quality synthesis when editing multiple attributes as illustrated in Fig.4– 5. These qualitative results also showcase the strong independence between the attributes. However, our contrastive approach fails to disentangle the *orientation* factor on DSprites, which degrades the overall performance. This may indicate that certain controls cannot be fully captured by a single dimension and enforcing compactness may also be counterproductive as argued in prior works (Carbonneau et al., 2022). Supporting this observation, while features independence is fully observed on 3DShapes, the color classes in Fig 4 do not seem to follow a clear ordering, which limits the smoothness of morphing results.

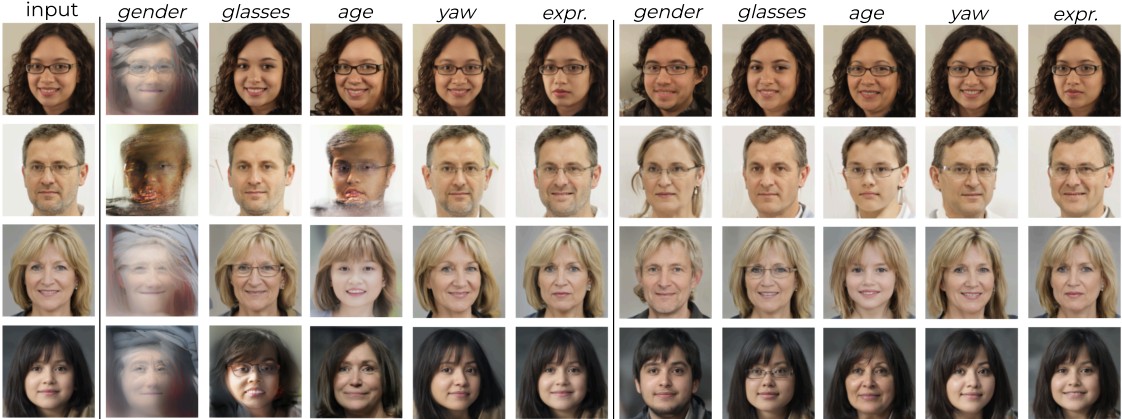

Figure 5: Attribute editing on FFHQ comparing PluGeN (*left*) against our iterative method M2 (*right*).

Table 5: Editing accuracy (%) for the targeted attribute on FFHQ.

| Edited | Ours ($c_1$) | Ours ($c_K$) |
|---|---|---|
| Gender | **92.30** | 90.60 |
| Glasses | **98.6** | 97.5 |
| Yaw | 32.15 | 32.15 |
| Pitch | **33.65** | 32.8 |
| Baldness | **97.45** | 97.25 |
| Facial Hair | **94.45** | 89.45 |
| Age | **41.45** | 38.60 |
| Expression | **86.65** | 82.35 |

Table 6: Control accuracy (%) and FID quality score for direct reconstruction and editing tasks on FFHQ. Target Acc. measures the accuracy of the edited attribute, while Other Acc. measures the mean accuracy over the remaining controls (detailed in the appendix A.3.2).

| Model | Target Acc.↑ | Other Acc.↑ | Mean Acc.↑ | FID↓ |
|---|---|---|---|---|
| Ours ($c_1$) - rec. | **77.30** | **76.68** | **76.76** | **20.691** |
| Ours ($c_K$) - rec. | 75.88 | 75.88 | 75.88 | 24.917 |
| Ours ($c_1$) - edit. | **72.09** | 74.87 | 74.53 | **22.819** |
| Ours ($c_K$) - edit. | 70.09 | **75.44** | **74.77** | 26.088 |

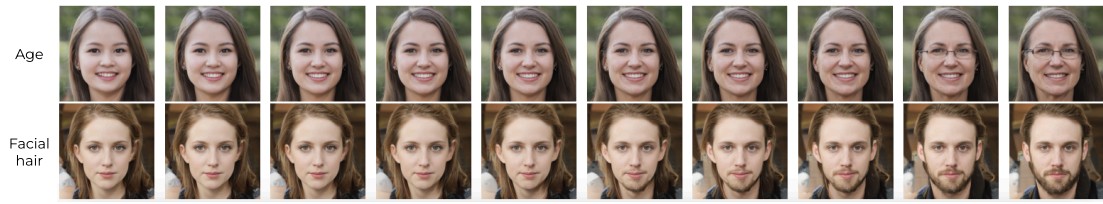

Figure 6: Traversals in the control space of single-control models (M2 with only $c_1$) on FFHQ.

When the targeted controls are successfully disentangled, we obtain smooth interpolations between features on the re-synthesized images, which transforms discrete attributes such as *gender* or *glasses* on FFHQ into fine-grained continuous controls. However, our method is sensitive to dataset biases. As shown in Tables 5–6, learning one model per control improves the target prediction accuracy but can degrade the accuracy of other correlated features. As illustrated in Fig. 6, learning *age* in $c_1$ may lead the model to associate older individuals with glasses, while similar correlations appear between *facial hair* and *gender*. Although learning *glasses* before *age* alleviate this issue, other cases remain unresolved, highlighting a limitation of iterative learning and suggesting that joint control learning may still be required to enforce some independence.

## 6 Conclusion

We proposed an efficient disentanglement method that regularizes VAEs with contrastive learning to iteratively add customized controls on pretrained generative models while preserving the synthesis quality. Future work will focus on extending our approach to other modalities and reducing the need for annotated data.

**Broader Impact Statement**

This work contributes to improving controllability in generative models by enabling targeted manipulation of customized control representations. Such capabilities can benefit applications in creative industries, scientific analysis, and data augmentation, where fine-grained control over generated outputs is desirable. Additionally, increased interpretability of latent spaces may help researchers better understand model behavior.

However, improved controllability of generative models may also facilitate misuse. In particular, these control methods could be used to create misleading or deceptive media, including edited images that alter perceived attributes of individuals. Furthermore, latent directions learned from real-world data may encode and amplify societal biases, potentially leading to outputs that reinforce harmful stereotypes when manipulated along certain attributes.

To mitigate these risks, we recommend systematically auditing latent directions for bias using diverse datasets. Manipulations involving sensitive attributes should be restricted or carefully validated, with safeguards such as access controls or human oversight in higher-risk settings. We emphasize clear documentation of the method's limitations, especially that latent directions reflect correlations rather than causal relationships, and discourage use in high-stakes decision-making contexts. Additionally, responsible dataset curation and transparency about data composition are important to reduce bias, and we encourage reporting failure cases to prevent overreliance. Where applicable, controlled or research-focused release of code and models can further help limit harmful misuse.

To conclude, while our method improves controllability and interpretability in generative models, its deployment should be accompanied by appropriate safeguards to prevent misuse and unintended societal harm.

**Acknowledgments**

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

## A   Appendix

### A.1   Algorithms

We detail the algorithms of our proposed method in Algorithm 1 and 2 for M1, and Algorithm 3 for M2. For reproducibility, the source code is also available on our GitHub [2] along with the pretrained models.

#### A.1.1   Method 1 (M1): disentangling multiple controls

As a pre-processing step, we gather the training data $\mathbf{x}$ into groups $\mathcal{G}_k$ that provide individual views of each control variable $c_k$. Each group corresponds to the set of examples $\mathbf{x}$ whose associated multi-labels $\mathbf{y} = (y_1, ..., y_K)$ are all identical except for the $y_k$ component which varies. For each $c_k$, we denote as $\mathcal{X}_k$ the set of all these groups.

---

**Algorithm 1** Training procedure for our method M1

1:  **Input:** $K$ number of controls to disentangle, the training set $\mathcal{X} = \{\mathcal{X}_k\}_{k=1}^{K}$ where $X_k$ is the set of all the groups $\mathcal{G}_k$ representing $c_k$, batch size of $N$ groups, encoder $E_{\mathbf{z}}$ of the pretrained generative model, control extractor $E_{c\phi}$ and latent projector $D_{c\theta}$, learning rate $\gamma$
2:  **for** each training step **do**
3:      Encode $(\mu_{\mathbf{z}}, \mathbf{\Sigma_z}) = E_{\mathbf{z}}(\mathbf{x})$
4:      Sample $\mathbf{z} \sim \mathcal{N}(\mu_{\mathbf{z}}, \mathbf{\Sigma_z})$
5:      Encode $(\mu_{\mathbf{c}}, \mathbf{\Sigma_c}) = E_{c\phi}(\mathbf{z})$
6:      $\mathcal{L}_{KL} = \mathcal{D}_{\mathrm{KL}}\big[\mathcal{N}(\mu_{\mathbf{c}}, \mathbf{\Sigma_c}) \parallel \mathcal{N}(\mathbf{0}, \mathbf{I})\big]$
7:      Sample $\mathbf{c} \sim \mathcal{N}(\mu_{\mathbf{c}}, \mathbf{\Sigma_c})$
8:      Decode $\hat{\mathbf{z}} = D_{c\theta}(\mathbf{c})$
9:      Compute $\mathcal{L}_{\mathrm{rec}} = \frac{1}{N \times |\mathcal{G}|} \sum_{i=1}^{N \times |\mathcal{G}|} (\mathbf{z}_i - \hat{\mathbf{z}}_i)^2$
10:     Initialize variation sets $\mathcal{V} := \{\}$
11:     **for** $k = 0$ to $K - 1$ **do**
12:         Initialize variation set $\mathcal{V}_k := \{\}$
13:         **for all** $\mathcal{G}_k \in \mathcal{X}_k$ **do**
14:             $\mathbf{v}_k = |\mathbf{c}_k^{(\neq\mathrm{ref})} - \mathbf{c}_k^{(\mathrm{ref})}|$
15:             $\mathcal{V}_k \leftarrow \mathcal{V}_k \cup \{\mathbf{v}_k\}$
16:         **end for**
17:         $\mathcal{V} \leftarrow \mathcal{V} \cup \mathcal{V}_k$
18:     **end for**
19:     **for** $k = 0$ to $K - 1$ **do**
20:         $\mathcal{Q}_k, \mathcal{K}_k^{+} \sim \mathrm{RandomSplit}(\mathcal{V}_k)$
21:         $\mathcal{K}_k^{-} \leftarrow \texttt{SampleNegativeSet}(K, \mathcal{V}, k, cdims)$
22:         $\mathcal{L}_k = \sum_{(\boldsymbol{v}_q, \boldsymbol{v}_p) \in (\mathcal{Q}_k, \mathcal{K}_k^+)} \log \frac{\exp(\mathrm{sim}(\boldsymbol{v}_q, \boldsymbol{v}_p)/\tau)}{\exp(\mathrm{sim}(\boldsymbol{v}_q, \boldsymbol{v}_p)/\tau) + \sum_{\boldsymbol{v}_n \in \mathcal{K}_k^-} \exp(\mathrm{sim}(\boldsymbol{v}_q, \boldsymbol{v}_n)/\tau)}$
23:     **end for**
24:     Compute $\mathcal{L}_{\mathrm{InfoNCE}} = -\frac{1}{K} \sum_{k=1}^{K} \mathcal{L}_k$
25:     Loss $\mathcal{L}_{\theta,\phi} = \mathcal{L}_{\mathrm{rec}} + \mathcal{L}_{\mathrm{InfoNCE}} + \beta \cdot \mathcal{L}_{KL}$
26:     Update $\theta \leftarrow \theta - \gamma\nabla_\theta\mathcal{L}_{\theta,\phi}$ and $\phi \leftarrow \phi - \gamma\nabla_\phi\mathcal{L}_{\theta,\phi}$
27:  **end for**
28:  **return** $E_{c\phi}, D_{c\theta}$

---

**Algorithm 2** Sampling Negative Sets for method M1

1:  **Inputs:** $K$ number of controls, $\mathcal{V}$ variation sets, $k$ the control dimension currently optimized, $cdims$ boolean whether to replace only the $K$ first dimensions in $\mathcal{V}_k$
2:  **Function** $\texttt{SampleNegativeSet}(K, \mathcal{V}, k, cdims)$
3:  Initialize variation set $\mathcal{K}_k^- := \{\}$
4:  n_samples $= \frac{|\mathcal{V}_k|}{K-1}$
5:  **for** $l = 0$ to $K - 1$ **do**
6:      **if** $l \neq k$ **then**
7:          $\mathcal{K}_l \sim \mathrm{RandomSample}(\mathcal{V}_l, \mathrm{n\_samples})$
8:          $\mathcal{K}_k^- \leftarrow \mathcal{K}_k^- \cup \mathcal{K}_l$
9:      **end if**
10: **end for**
11: **if** cdims is True **then**
12:     $\mathcal{K}^- \leftarrow \texttt{clone}(\mathcal{V}_k)$
13:     $\mathcal{K}^-[:,:K] = \mathcal{K}_k^-[:,:K]$
14:     $\mathcal{K}_k^- = \mathcal{K}^-$
15: **end if**
16: **return** $\mathcal{K}_k^-$

---

[2]https://github.com/anonymous/discovae (The GitHub link will be updated upon acceptance notification.)

### A.1.2 Method 2 (M2): disentangling one control at a time

---

**Algorithm 3** Training procedure for our method M2

---

1: **Input:** $k$ the current control dimension to disentangle, batch of N examples $\mathcal{X} = \{\mathbf{x}_n\}_{n=1}^N$ paired with labels $\mathcal{Y} = \{y_n\}_{n=1}^N$, with $y_n \in [\![1, M_k]\!]$, $\mathcal{A} \subseteq [\![1, M_k]\!]$ the subset of available class labels in the current batch, encoder $E_{\mathbf{z}}$ of the pretrained generative model, pretrained control extractors $E_{c<k}$, current control extractor $E_{c \geq k \phi}$ and latent projector $D_{c\theta}$, learning rate $\gamma$
2: **for** each training step **do**
3:     Encode $(\mu_{\mathbf{z}}, \Sigma_{\mathbf{z}}) = E_{\mathbf{z}}(\mathbf{x})$
4:     Sample $\mathbf{z} \sim \mathcal{N}(\mu_{\mathbf{z}}, \Sigma_{\mathbf{z}})$
5:     Encode $(\mu_{\mathbf{c}}, \Sigma_{\mathbf{c}}) = E_{\mathbf{c} \geq \mathbf{k}\phi}(\mathbf{z})$
6:     $\mathcal{L}_{KL} = \mathcal{D}_{\mathrm{KL}}\big[\mathcal{N}(\mu_{\mathbf{c}}, \Sigma_{\mathbf{c}}) \,\|\, \mathcal{N}(\mathbf{0}, \mathbf{I})\big]$
7:     Sample $\mathbf{c}_{\geq k} \sim \mathcal{N}(\mu_{\mathbf{c}}, \Sigma_{\mathbf{c}})$
8:     Encode $(\mu_{\mathbf{c} < \mathbf{k}}, \Sigma_{\mathbf{c} < \mathbf{k}}) = E_{\mathbf{c} < \mathbf{k}}(\mathbf{z})$
9:     $\mathbf{c}_{<k} \leftarrow \mu_{\mathbf{c} < \mathbf{k}}$
10:     $\mathbf{c}_{\geq k} \leftarrow \mathbf{c}_{\geq k}[:, k :]$
11:     Compute $\mathbf{c} = \texttt{concatenate}(\mathbf{c}_{<k}, \mathbf{c}_{\geq k})$
12:     Decode $\hat{\mathbf{z}} = D_{\mathbf{c}\theta}(\mathbf{c})$
13:     Compute $\mathcal{L}_{\mathrm{rec}} = \frac{1}{N} \sum_{n=1}^N (\mathbf{z}_n - \hat{\mathbf{z}}_n)^2$
14:     Randomly partition $\mathcal{C} = \{\mathbf{c}_n\}_{n=1}^N$ into two disjoint subsets $\mathcal{C}^Q$ and $\mathcal{C}^+$ such that each class is equally represented
15:     Initialize variation sets $\mathcal{Q} = \{\}$, $\mathcal{K}^+ = \{\}$, $\mathcal{K}^- = \{\}$
16:     **for** each sample $(\boldsymbol{c}_{i_1}, \boldsymbol{c}_{i_2}) \in \mathcal{C}^Q \times \mathcal{C}^+$ such that $y_{i_1} = y_{i_2} = i \in \mathcal{A}$ {positive pair} **do**
17:         Randomly sample $(\boldsymbol{c}_{j_1}, \boldsymbol{c}_{j_2}) \in \mathcal{C}^Q \times \mathcal{C}^+$ such that $y_{j_1} \neq i$ and $y_{j_2} \neq i$ {negative pair}
18:         Compute $\boldsymbol{v}_q = |\boldsymbol{c}_{i_1} - \boldsymbol{c}_{j_1}|$ and $\boldsymbol{v}_p = |\boldsymbol{c}_{i_2} - \boldsymbol{c}_{j_2}|$
19:         $\mathcal{Q} \leftarrow \mathcal{Q} \cup \{\boldsymbol{v}_q\}$ and $\mathcal{K}^+ \leftarrow \mathcal{K}^+ \cup \{\boldsymbol{v}_p\}$
20:         $\boldsymbol{c}_p = \texttt{concatenate}(\boldsymbol{c}_{i_2}[:, k], \boldsymbol{c}_{j_2}[:, \neq k])$
21:         $\boldsymbol{v}_{n_1} = |\boldsymbol{c}_{i_1} - \boldsymbol{c}_p|$
22:         $\boldsymbol{c}_q = \texttt{concatenate}(\boldsymbol{c}_{i_1}[:, k], \boldsymbol{c}_{j_1}[:, \neq k])$
23:         $\boldsymbol{v}_{n_2} = |\boldsymbol{c}_{i_2} - \boldsymbol{c}_q|$
24:         $\mathcal{K}^- \leftarrow \mathcal{K}^- \cup \{\boldsymbol{v}_{n_1}, \boldsymbol{v}_{n_2}\}$
25:     **end for**
26:     $\mathcal{L}_{\mathrm{InfoNCE}} = \frac{1}{|\mathcal{Q}|} \sum_{(\boldsymbol{v}_q, \boldsymbol{v}_p) \in (\mathcal{Q}, \mathcal{K}^+)} \log \frac{\exp(\mathrm{sim}(\boldsymbol{v}_q, \boldsymbol{v}_p)/\tau)}{\exp(\mathrm{sim}(\boldsymbol{v}_q, \boldsymbol{v}_p)/\tau) + \sum_{\boldsymbol{v}_n \in \mathcal{K}^-} \exp(\mathrm{sim}(\boldsymbol{v}_q, \boldsymbol{v}_n)/\tau)}$
27:     Loss $\mathcal{L}_{\theta, \phi} = \mathcal{L}_{\mathrm{rec}} + \mathcal{L}_{\mathrm{InfoNCE}} + \beta \cdot \mathcal{L}_{KL}$
28:     Update $\theta \leftarrow \theta - \gamma \nabla_\theta \mathcal{L}_{\theta, \phi}$ and $\phi \leftarrow \phi - \gamma \nabla_\phi \mathcal{L}_{\theta, \phi}$
29: **end for**
30: **return** $E_{c\phi}, D_{c\theta}$

---

### A.2 Implementation details

#### A.2.1 Architectures of our VAE-based method

We detail the network's architecture of our VAE-based model in Table 7 for the *control extractor* and in Table 8 for the *latent projector*. The architecture is similar to the one used in TwoStageVAE (Dai & Wipf, 2019). $N$ corresponds to the batch size and $D_z$ is the number of latent dimensions, which is equal to 10, 64 and 512 for DSprites, 3DShapes and FFHQ, respectively.

Table 7: Network's architecture of the *Control Extractor $E_{\mathbf{c}}$*.

| Layer | Operation | Output Shape |
|---|---|---|
| Input | Latent vector $z$ | $(N, D_z)$ |
| FC 1 | Linear$(D_z \to 1024)$ + ReLU | $(N, 1024)$ |
| FC 2 | Linear$(1024 \to 1024)$ + ReLU | $(N, 1024)$ |
| FC 3 | Linear$(1024 \to 1024)$ + ReLU | $(N, 1024)$ |
| Concat | Concatenate input $z$ and last layer | $(N, 1024 + D_z)$ |
| $\mu$ | Linear$(1024 + D_z \to D_z)$ | $(N, D_z)$ |
| $\log \sigma^2$ | Linear$(1024 + D_z \to D_z)$ | $(N, D_z)$ |

Table 8: Network's architecture of the *Latent Projector $D_{\mathbf{c}}$*.

| Layer | Operation | Output Shape |
|---|---|---|
| Input | Control vector $c$ | $(N, D_z)$ |
| FC 1 | Linear$(D_z \to 1024)$ + ReLU | $(N, 1024)$ |
| FC 2 | Linear$(1024 \to 1024)$ + ReLU | $(N, 1024)$ |
| FC 3 | Linear$(1024 \to 1024)$ + ReLU | $(N, 1024)$ |
| Concat | Concatenate input $c$ and last layer | $(N, 1024 + D_z)$ |
| $z$ | Linear$(1024 + D_z \to D_z)$ | $(N, D_z)$ |

#### A.2.2 Pretrained VAE

We pre-trained convolutional VAEs on DSprites and 3DShapes for 1M and 300k steps, respectively, using $\beta = 1$ with 10 and 64 latent dimensions and architectures similar to DisCo's encoders that are described in Table 9 and Table 10 for DSprites, and Table 11 and Table 12 for 3DShapes.

Table 9: Architecture of the *Encoder* of the $\beta$-VAE on DSprites.

| Layer | Operation | Output Shape |
|---|---|---|
| Input | Grayscale image | $(N, 1, 64, 64)$ |
| Conv 1 | Conv2d$(1 \to 32$, k=4, stride=2, pad=1) + ReLU | $(N, 32, 32, 32)$ |
| Conv 2 | Conv2d$(32 \to 32$, k=4, stride=2, pad=1) + ReLU | $(N, 32, 16, 16)$ |
| Conv 3 | Conv2d$(32 \to 32$, k=4, stride=2, pad=1) + ReLU | $(N, 32, 8, 8)$ |
| Conv 4 | Conv2d$(32 \to 32$, k=4, stride=2, pad=1) + ReLU | $(N, 32, 4, 4)$ |
| Flatten | Flatten | $(N, 512)$ |
| FC 1 | Linear$(512 \to 256)$ + ReLU | $(N, 256)$ |
| FC 2 | Linear$(256 \to 256)$ + ReLU | $(N, 256)$ |
| $\mu, \log \sigma^2$ | Linear$(256 \to 2\times 10)$ | $(N, 2 \times 10)$ |

Table 10: Architecture of the *Decoder* of the $\beta$-VAE on DSprites.

| Layer | Operation | Output Shape |
|---|---|---|
| Input | Latent vector $z$ | $(N, 10)$ |
| FC 1 | Linear($10 \to 256$) + ReLU | $(N, 256)$ |
| FC 2 | Linear($256 \to 256$) + ReLU | $(N, 256)$ |
| FC 3 | Linear($256 \to 512$) + ReLU | $(N, 512)$ |
| Reshape | Reshape to feature map | $(N, 32, 4, 4)$ |
| Deconv 1 | ConvTranspose2d($32 \to 32$, k=4, stride=2, pad=1) + ReLU | $(N, 32, 8, 8)$ |
| Deconv 2 | ConvTranspose2d($32 \to 32$, k=4, stride=2, pad=1) + ReLU | $(N, 32, 16, 16)$ |
| Deconv 3 | ConvTranspose2d($32 \to 32$, k=4, stride=2, pad=1) + ReLU | $(N, 32, 32, 32)$ |
| Deconv 4 | ConvTranspose2d($32 \to 1$, k=4, stride=2, pad=1) + Sigmoid | $(N, 1, 64, 64)$ |

Table 11: Architecture of the *Encoder* of the $\beta$-VAE on 3DShapes.

| Layer | Operation | Output Shape |
|---|---|---|
| Input | RGB image | $(N, 3, 64, 64)$ |
| Conv 1 | Conv2d($3 \to 64$, k=7, stride=1, pad=3) + LeakyReLU(0.01) | $(N, 64, 64, 64)$ |
| Conv 2 | Conv2d($64 \to 128$, k=4, stride=2, pad=1) + LeakyReLU(0.01) | $(N, 128, 32, 32)$ |
| Conv 3 | Conv2d($128 \to 256$, k=4, stride=2, pad=1) + LeakyReLU(0.01) | $(N, 256, 16, 16)$ |
| Conv 4 | Conv2d($256 \to 256$, k=4, stride=2, pad=1) + LeakyReLU(0.01) | $(N, 256, 8, 8)$ |
| Conv 5 | Conv2d($256 \to 256$, k=4, stride=2, pad=1) + LeakyReLU(0.01) | $(N, 256, 4, 4)$ |
| Flatten | Flatten | $(N, 4096)$ |
| FC 1 | Linear($4096 \to 256$) + LeakyReLU(0.01) | $(N, 256)$ |
| FC 2 | Linear($256 \to 256$) + LeakyReLU(0.01) | $(N, 256)$ |
| $\mu, \log \sigma^2$ | Linear($256 \to 2{\times}64$) | $(N, 2 \times 64)$ |

.

Table 12: Architecture of the *Decoder* of the $\beta$-VAE on 3DShapes.

| Layer | Operation | Output Shape |
|---|---|---|
| Input | Latent vector $z$ | $(N, 64)$ |
| FC 1 | Linear($64 \to 256$) + ReLU | $(N, 256)$ |
| FC 2 | Linear($256 \to 256$) + ReLU | $(N, 256)$ |
| FC 3 | Linear($256 \to 4096$) + ReLU | $(N, 4096)$ |
| Reshape | Reshape to feature map | $(N, 256, 4, 4)$ |
| ConvT 1 | ConvTranspose2d($256 \to 256$, k=4, stride=2, pad=1) + ReLU | $(N, 256, 8, 8)$ |
| ConvT 2 | ConvTranspose2d($256 \to 256$, k=4, stride=2, pad=1) + ReLU | $(N, 256, 16, 16)$ |
| ConvT 3 | ConvTranspose2d($256 \to 128$, k=4, stride=2, pad=1) + ReLU | $(N, 128, 32, 32)$ |
| ConvT 4 | ConvTranspose2d($128 \to 64$, k=4, stride=2, pad=1) + ReLU | $(N, 64, 64, 64)$ |
| ConvT 5 | ConvTranspose2d($64 \to 3$, k=7, stride=1, pad=3) | $(N, 3, 64, 64)$ |

### A.2.3 Pretrained classifier

To evaluate the consistency of re-synthesized images under feature modifications, we pretrained multi-head classifiers on each dataset for 20 epochs. We detail the network architecture and control prediction accuracy results on the test set in Table 13 and Table 14 for DSprites, and Table 15 and Table 16 for 3DShapes. For FFHQ, we pre-compute CLIP embeddings using the `open-clip` library [3] and train only the projection heads as described in Table 17. We provide the accuracy prediction results for each attributes on the test set in Table 18.

Table 13: Architecture of the *MultiHeadClassifier* network for DSprites.

| Layer | Operation | Output Shape |
|---|---|---|
| | *Encoder* | |
| Input | Grayscale image | $(\cdot, 1, 64, 64)$ |
| Conv 1 | Conv2d($1 \to 16$, k=4, stride=2, pad=1) + ReLU | $(\cdot, 16, 32, 32)$ |
| Conv 2 | Conv2d($16 \to 32$, k=4, stride=2, pad=1) + ReLU | $(\cdot, 32, 16, 16)$ |
| Conv 3 | Conv2d($32 \to 64$, k=4, stride=2, pad=1) + ReLU | $(\cdot, 64, 8, 8)$ |
| Flatten | Flatten | $(\cdot, 4096)$ |
| | *Multi-head Projection Heads* | |
| Head 1 | Linear($4096 \to 3$) | $(\cdot, 3)$ |
| Head 2 | Linear($4096 \to 6$) | $(\cdot, 6)$ |
| Head 3 | Linear($4096 \to 40$) | $(\cdot, 40)$ |
| Head 4 | Linear($4096 \to 32$) | $(\cdot, 32)$ |
| Head 5 | Linear($4096 \to 32$) | $(\cdot, 32)$ |

Table 14: Evaluation of the pretrained classifier on the test set. Accuracy per factor on DSprites dataset.

| Attribute | Accuracy (%) |
|---|---|
| Shape | 100.00 |
| Scale | 99.67 |
| Orientation | 86.29 |
| Position X | 98.99 |
| Position Y | 98.75 |

---

[3]https://github.com/mlfoundations/open_clip.git

Table 15: Architecture of the *MultiHeadClassifier* network for 3DShapes.

| Layer | Operation | Output Shape |
|---|---|---|
| | *Encoder* | |
| Input | RGB image | $(\cdot, 3, 64, 64)$ |
| Conv 1 | Conv2d($3 \to 16$, k=4, stride=2, pad=1) + ReLU | $(\cdot, 16, 32, 32)$ |
| Conv 2 | Conv2d($16 \to 32$, k=4, stride=2, pad=1) + ReLU | $(\cdot, 32, 16, 16)$ |
| Conv 3 | Conv2d($32 \to 64$, k=4, stride=2, pad=1) + ReLU | $(\cdot, 64, 8, 8)$ |
| Flatten | Flatten | $(\cdot, 4096)$ |
| | *Multi-head Projection Heads* | |
| Head 1 | Linear($4096 \to 10$) | $(\cdot, 10)$ |
| Head 2 | Linear($4096 \to 10$) | $(\cdot, 10)$ |
| Head 3 | Linear($4096 \to 10$) | $(\cdot, 10)$ |
| Head 4 | Linear($4096 \to 8$) | $(\cdot, 8)$ |
| Head 5 | Linear($4096 \to 4$) | $(\cdot, 4)$ |
| Head 6 | Linear($4096 \to 15$) | $(\cdot, 15)$ |

Table 16: Evaluation of the pretrained classifier on the test set. Accuracy per factor on 3DShapes dataset.

| Attribute | Accuracy (%) |
|---|---|
| Floor Color | 100.00 |
| Wall Color | 100.00 |
| Object Color | 100.00 |
| Object Size | 99.95 |
| Object Type | 100.00 |
| Orientation | 100.00 |

Table 17: Architecture of the *MultiHeadClassifier* with 8 separate heads.

| Head | Operation | Output |
|---|---|---|
| | *MultiHeadClassifier* | |
| Head 0 | Linear($512 \to 128$)+ ReLU + Dropout(p=0.4) + Linear($128 \to 2$) | $(\cdot, 2)$ |
| Head 1 | Linear($512 \to 128$) + ReLU + Dropout(p=0.4) + Linear($128 \to 2$) | $(\cdot, 2)$ |
| Head 2 | Linear($512 \to 128$) + ReLU + Dropout(p=0.4) + Linear($128 \to 6$) | $(\cdot, 6)$ |
| Head 3 | Linear($512 \to 128$) + ReLU + Dropout(p=0.4) + Linear($128 \to 6$) | $(\cdot, 6)$ |
| Head 4 | Linear($512 \to 128$) + ReLU + Dropout(p=0.4) + Linear($128 \to 2$) | $(\cdot, 2)$ |
| Head 5 | Linear($512 \to 128$) + ReLU + Dropout(p=0.4) + Linear($128 \to 2$) | $(\cdot, 2)$ |
| Head 6 | Linear($512 \to 128$) + ReLU + Dropout(p=0.4) + Linear($128 \to 8$) | $(\cdot, 8)$ |
| Head 7 | Linear($512 \to 128$) + ReLU + Dropout(p=0.4) + Linear($128 \to 2$) | $(\cdot, 2)$ |

Table 18: Evaluation of the pretrained classifier on the test set. Accuracy per factor on the FFHQ dataset.

| Attribute | Accuracy (%) |
|---|---|
| Gender | 96.15 |
| Glasses | 98.95 |
| Yaw | 43.50 |
| Pitch | 44.05 |
| Baldness | 98.75 |
| Beard | 97.45 |
| Age | 66.75 |
| Expression | 94.40 |

.

### A.3 Additional results

### A.3.1 Conditional synthesis

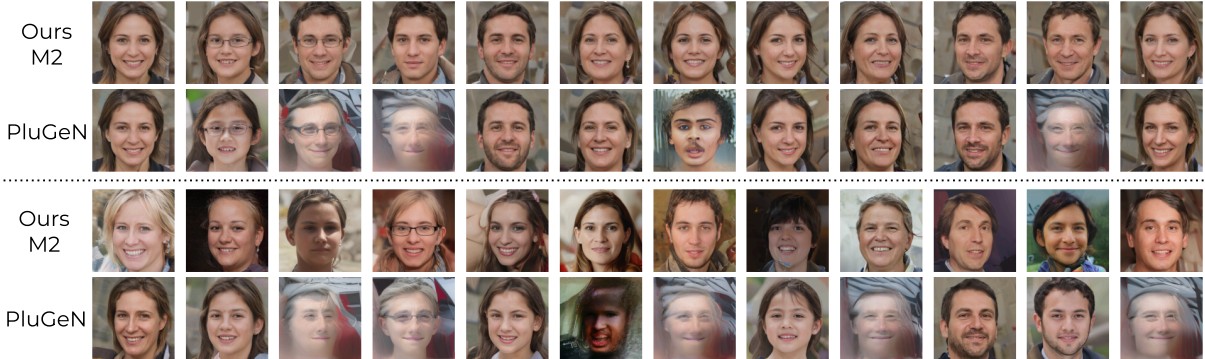

Figure 7: Examples of generated images with our iterative method M2 and with the PluGeN baseline trained on FFHQ for the conditional synthesis task where we directly sample random combinations of the Gaussian mixtures for selecting control attributes and we randomly sample the remaining latent features from an isotropic Gaussian distribution $\mathcal{N}(\mathbf{0}, \mathbf{I})$. On the *top*, we keep the same random vector for the remaining latent factors and only randomly change the attributes combinations, while on the *bottom*, we sample a different random vector for each image. Unlike PluGeN, our method maintains the synthesis quality regardless of the combination of attributes.

### A.3.2 Attribute editing with our method M2 on FFHQ: single-control models vs multi-control model

Comparing Table 19 with Table 20, we especially observe that the correlation between *glasses* and *age* when using single-control models is removed in the multi-control model. However, the correlation between *gender* and *facial hair* still remains. In particular, the multi-control model fails to completely decorrelate the *facial hair* attribute in the specified dimension. As illustrated in the traversals below, this control dimension only influences attribute editing in images of men, but has no effect on images of women. One might expect to observe a similar correlation effect for the *baldness* attribute, particularly with *gender* and *age*. However, we observe that the model simply fails to learn this control, even when using the single-control model. We believe this may be explained by the limited amount of annotated data corresponding to the absence of hair, which creates a strong class imbalance. As a result, the model does not observe enough variation for this control during training. As shown in the traversals below, PluGeN also suffers from this lack of annotations.

Table 19: Control accuracy (%) of single-control models when editing a specific attribute. Each column corresponds to the attribute being edited. The last column reports the mean accuracy on reconstructed images (no editing). We underline the scores of the other attributes impacted by editing the attribute, indicating correlations between these features.

|  | Gender | Glasses | Age | Yaw | Pitch | Bald. | Fac. Hair | Expr. | Rec. Mean |
|---|---|---|---|---|---|---|---|---|---|
| Gender | **92.30** | 93.95 | 89.85 | 94.05 | 95.05 | 93.05 | 79.75 | 93.55 | 95.02 |
| Glasses | 97.85 | **98.60** | 89.35 | 98.55 | 97.70 | 98.35 | 97.40 | 98.50 | 98.60 |
| Age | 48.45 | 50.30 | **41.45** | 56.55 | 56.40 | 54.80 | 51.65 | 57.00 | 58.66 |
| Yaw | 39.45 | 39.85 | 39.80 | **32.15** | 38.30 | 40.75 | 38.75 | 40.00 | 40.44 |
| Pitch | 38.45 | 36.40 | 35.45 | 37.85 | **33.65** | 37.10 | 36.40 | 36.30 | 38.74 |
| Baldness | 98.10 | 98.65 | 97.75 | 98.60 | 98.55 | **97.45** | 97.90 | 98.65 | 98.44 |
| Facial Hair | 89.35 | 96.60 | 94.95 | 96.15 | 96.85 | 95.15 | **94.45** | 95.30 | 96.17 |
| Expression | 86.75 | 91.05 | 86.50 | 85.85 | 88.85 | 88.25 | 86.15 | **86.65** | 88.91 |
| **Mean** | 73.84 | 75.68 | 71.89 | 74.97 | 75.67 | 75.61 | 72.81 | 75.74 | 76.71 |

Table 20: Control accuracy (%) our multi-control model when editing a specific attribute using a single model trained with all controls added iteratively. Each column corresponds to the attribute being edited. The last column reports the mean accuracy on reconstructed images (no editing). We underline the scores of the other attributes impacted by editing the attribute, indicating correlations between these features.

|  | Gender | Glasses | Age | Yaw | Pitch | Bald. | Fac. Hair | Expr. | Rec. Mean |
|---|---|---|---|---|---|---|---|---|---|
| Gender | **90.60** | 93.45 | 92.40 | 93.90 | 93.70 | 93.70 | 93.60 | 93.85 | 93.70 |
| Glasses | 97.70 | **97.50** | 97.75 | 97.70 | 97.65 | 97.55 | 97.65 | 97.60 | 97.60 |
| Age | 51.70 | 53.20 | **38.60** | 53.40 | 53.75 | 54.65 | 54.35 | 54.55 | 55.00 |
| Yaw | 37.70 | 38.45 | 39.50 | **32.15** | 38.45 | 39.25 | 39.00 | 39.10 | 38.75 |
| Pitch | 36.35 | 37.40 | 37.65 | 37.35 | **32.80** | 37.55 | 37.65 | 37.40 | 37.75 |
| Baldness | 98.05 | 98.25 | 97.85 | 98.30 | 98.25 | **97.25** | 98.00 | 98.40 | 98.35 |
| Facial Hair | 89.45 | 96.25 | 96.05 | 96.30 | 96.30 | 96.20 | **89.45** | 96.15 | 96.20 |
| Expression | 88.50 | 90.25 | 88.50 | 88.30 | 89.80 | 89.95 | 89.15 | **82.35** | 89.65 |
| **Mean** | 73.76 | 75.59 | 73.54 | 74.68 | 75.09 | 75.76 | 74.86 | 74.93 | 75.88 |

### A.3.3 Traversals on FFHQ

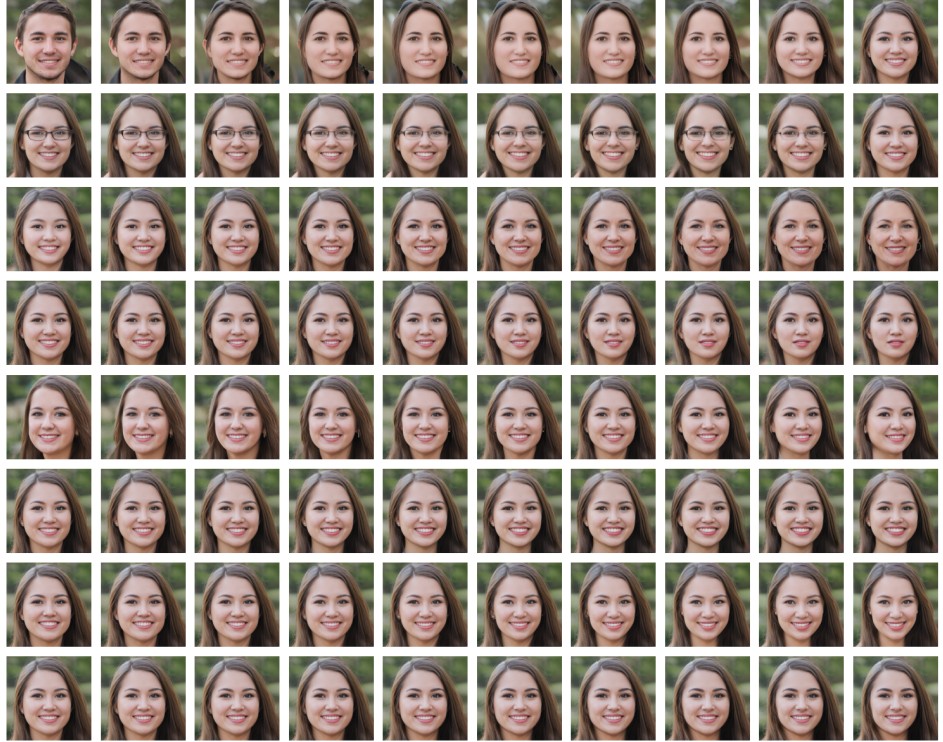

Figure 8: Traversals in the control space of our method M2 with one model for $K$ controls added iteratively: (1)*gender*, (2)*glasses*, (3)*age*, (4)*expression*, (5)*yaw*, (6)*baldness*, (7)*pitch*, (8)*facial hair*.

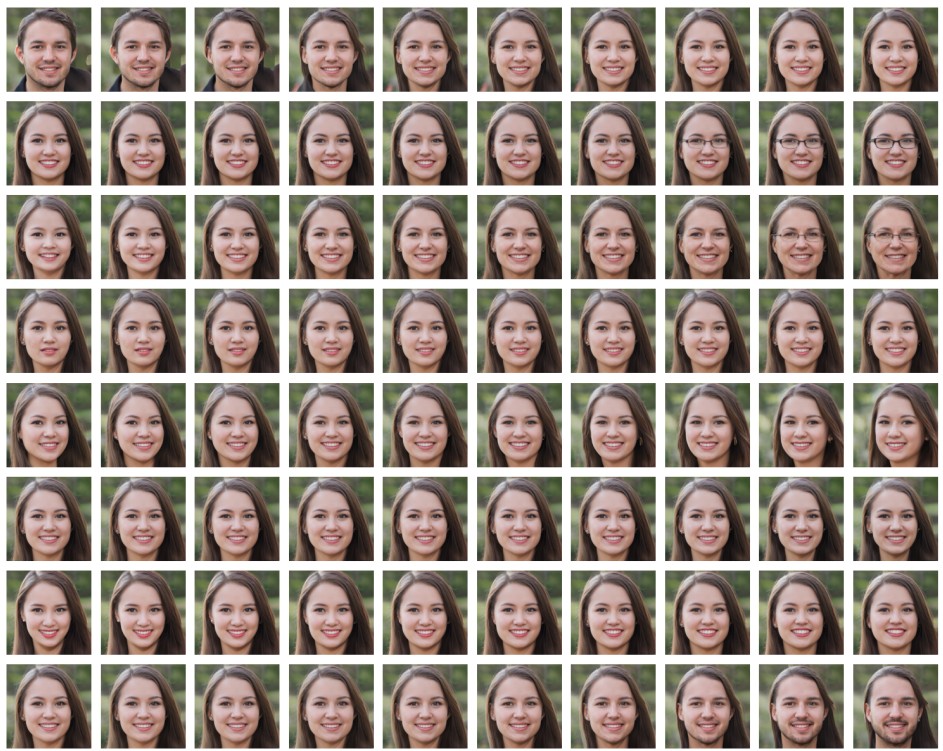

Figure 9: Traversals in the control space of our method M2 that disentangles a single labeled feature, which yields $K$ models for: $(1) gender$, $(2) glasses$, $(3) age$, $(4) expression$, $(5) yaw$, $(6) baldness$, $(7) pitch$, $(8) facial\ hair$.

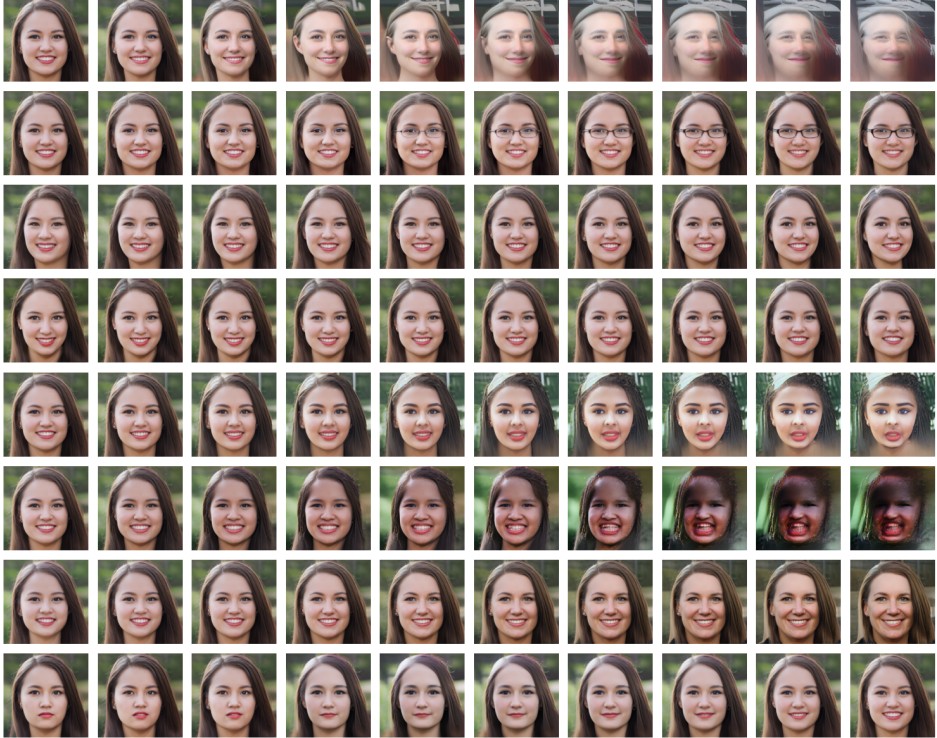

Figure 10: Traversals in the control space of PluGeN with $K$ controls: $(1) gender$, $(2) glasses$, $(3) yaw$, $(4) pitch$, $(5) baldness$, $(6) facial\ hair$, $(7) age$, $(8) expression$.

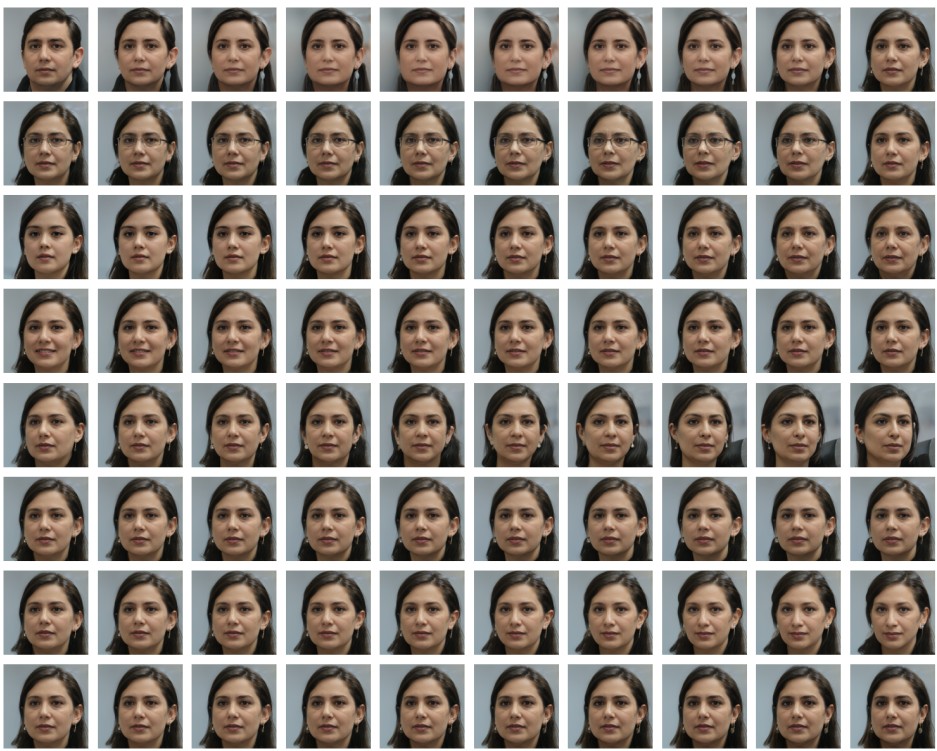

Figure 11: Traversals in the control space of our method M2 with one model for $K$ controls added iteratively: (1)*gender*, (2)*glasses*, (3)*age*, (4)*expression*, (5)*yaw*, (6)*baldness*, (7)*pitch*, (8)*facial hair*.

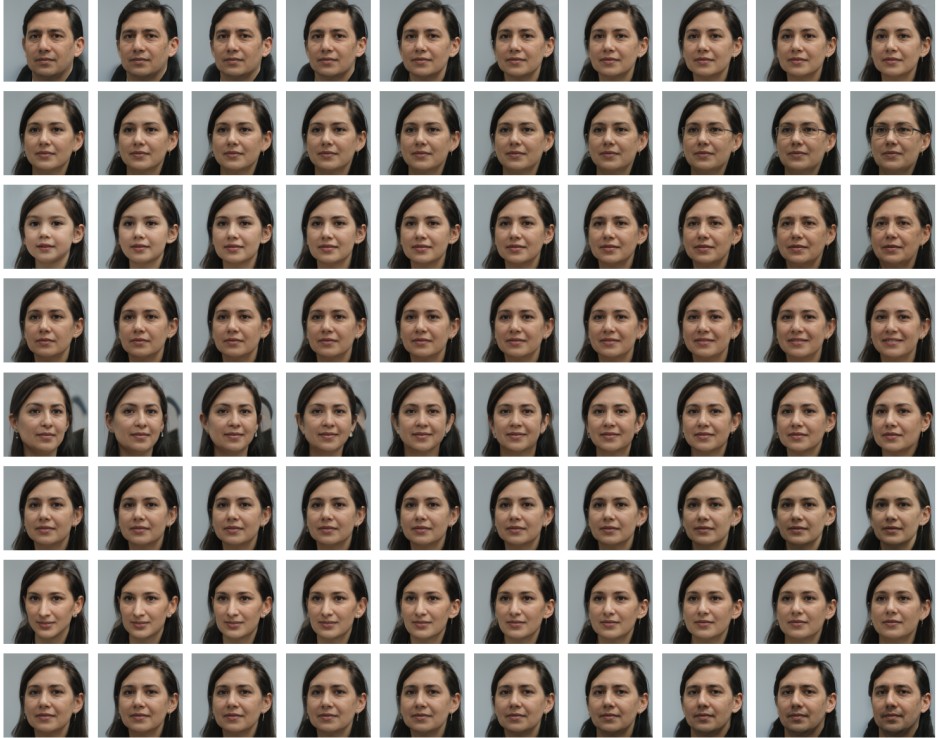

Figure 12: Traversals in the control space of our method M2 that disentangles a single labeled feature, which yields $K$ models for: (1)*gender*, (2)*glasses*, (3)*age*, (4)*expression*, (5)*yaw*, (6)*baldness*, (7)*pitch*, (8)*facial hair*.

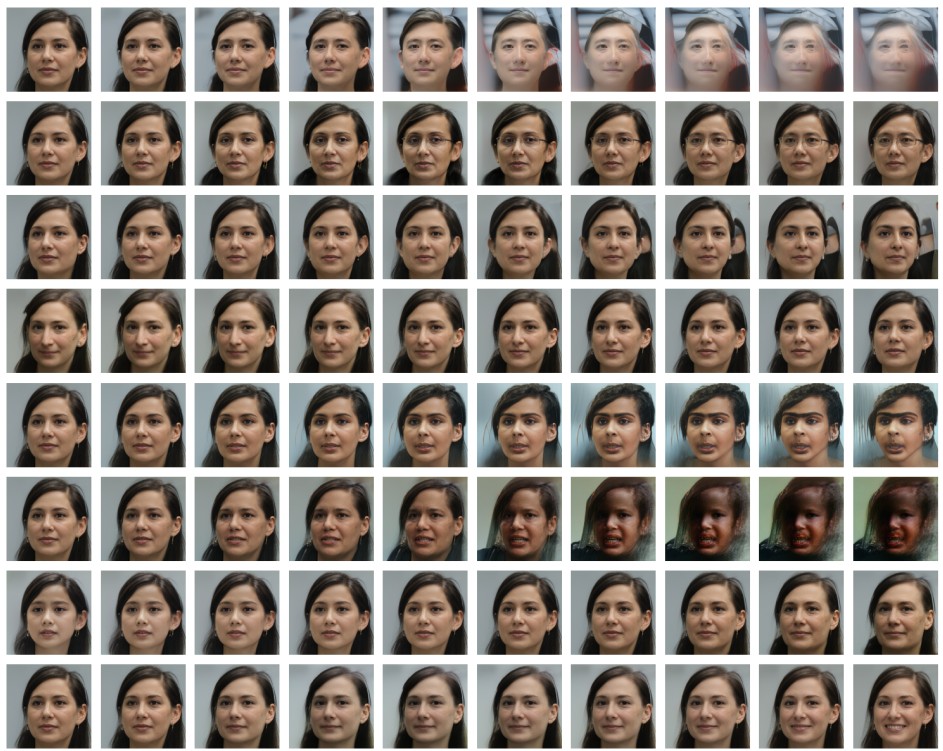

Figure 13: Traversals in the control space of PluGeN with $K$ controls: (1)*gender*, (2)*glasses*, (3)*yaw*, (4)*pitch*, (5)*baldness*, (6)*facial hair*, (7)*age*, (8)*expression*.

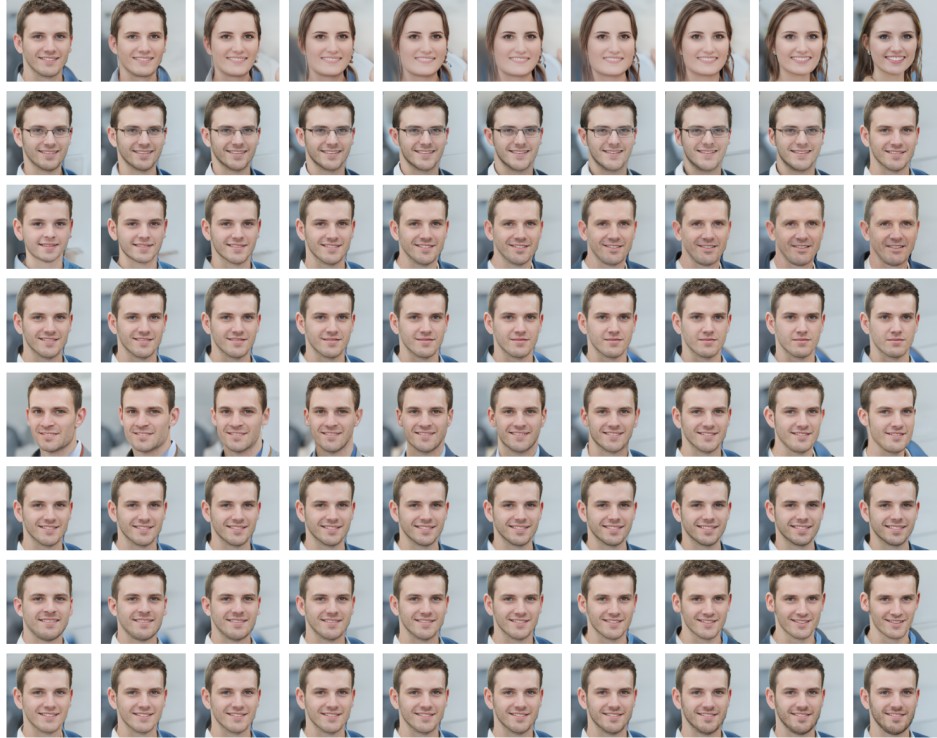

Figure 14: Traversals in the control space of our method M2 with one model for $K$ controls added iteratively: (1)*gender*, (2)*glasses*, (3)*age*, (4)*expression*, (5)*yaw*, (6)*baldness*, (7)*pitch*, (8)*facial hair*.

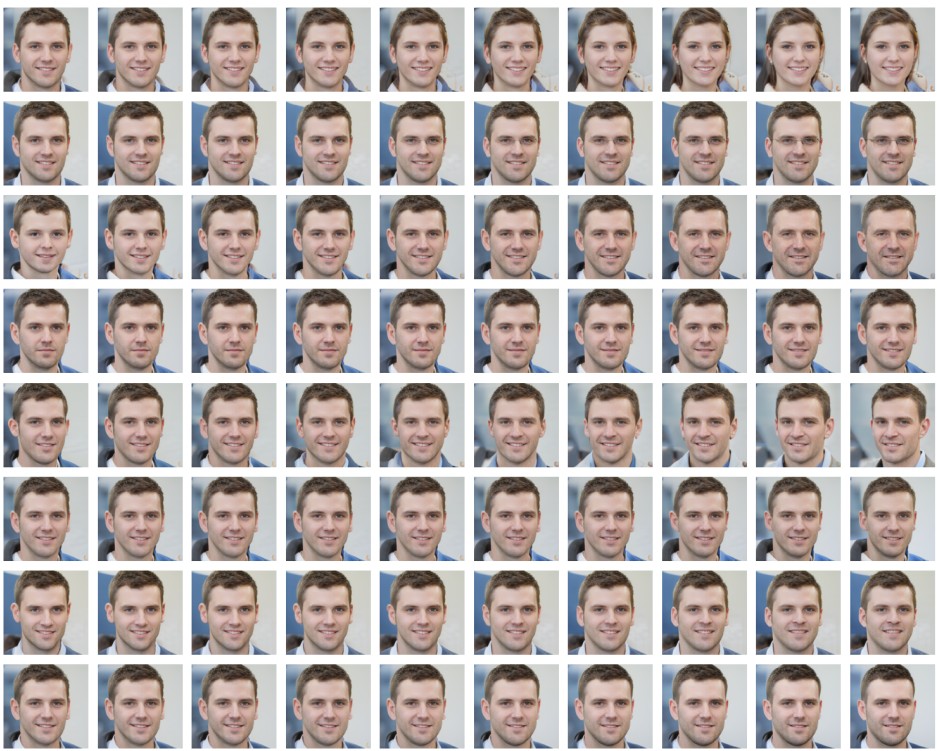

Figure 15: Traversals in the control space of our method M2 that disentangles a single labeled feature, which yields $K$ models for: (1)*gender*, (2)*glasses*, (3)*age*, (4)*expression*, (5)*yaw*, (6)*baldness*, (7)*pitch*, (8)*facial hair*.

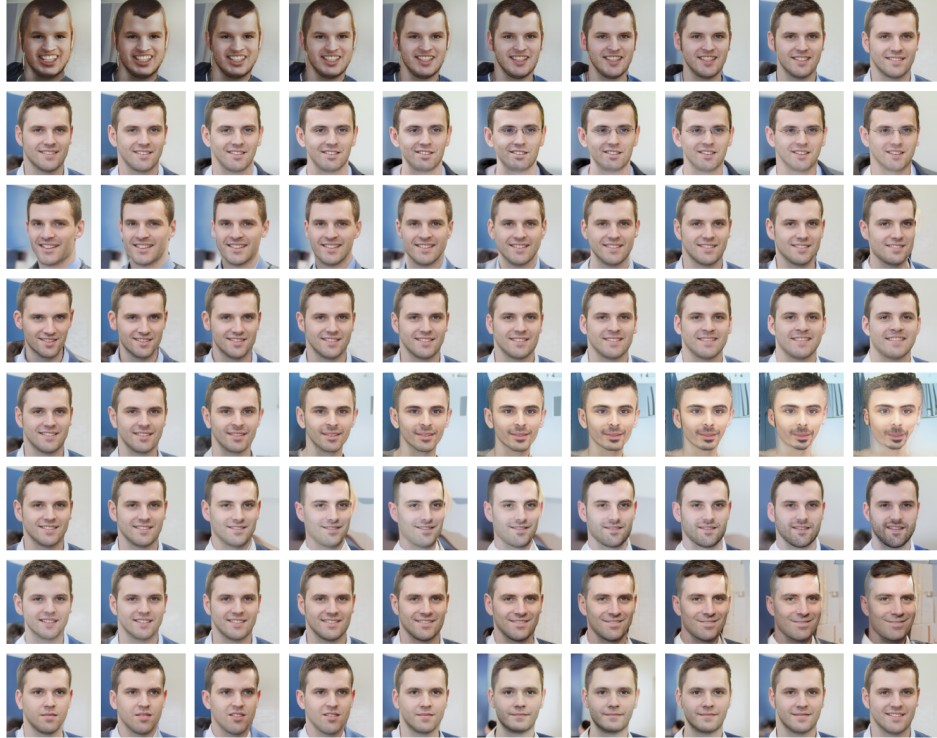

Figure 16: Traversals in the control space of PluGeN with $K$ controls: (1)*gender*, (2)*glasses*, (3)*yaw*, (4)*pitch*, (5)*baldness*, (6)*facial hair*, (7)*age*, (8)*expression*.

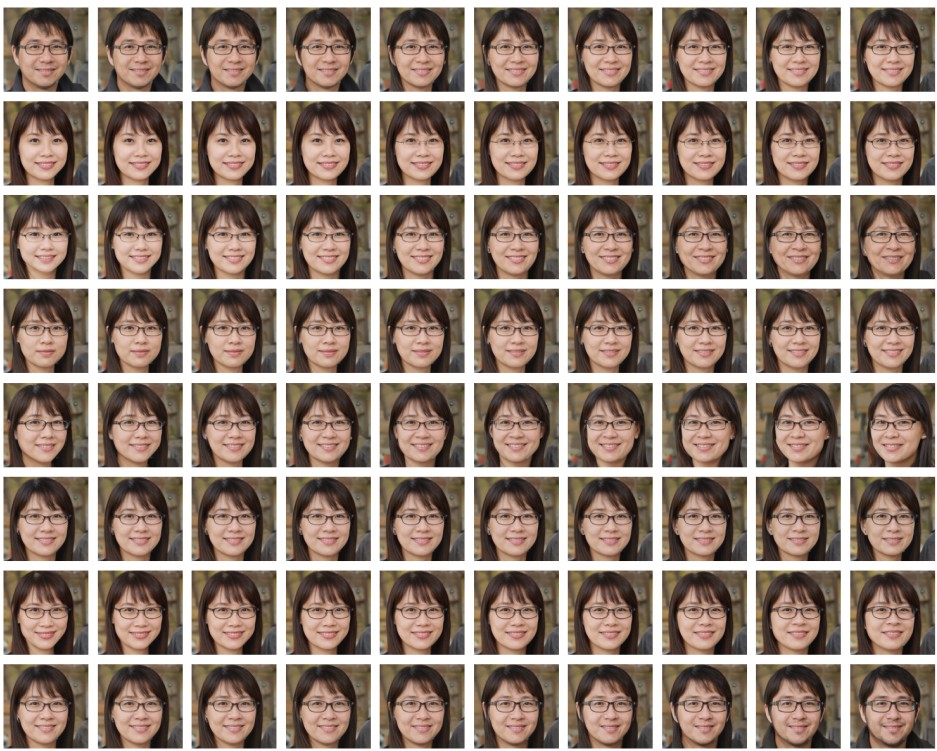

Figure 17: Traversals in the control space of our method M2 that disentangles a single labeled feature, which yields $K$ models for: $(1)gender$, $(2)glasses$, $(3)age$, $(4)expression$, $(5)yaw$, $(6)baldness$, $(7)pitch$, $(8)facial\ hair$.

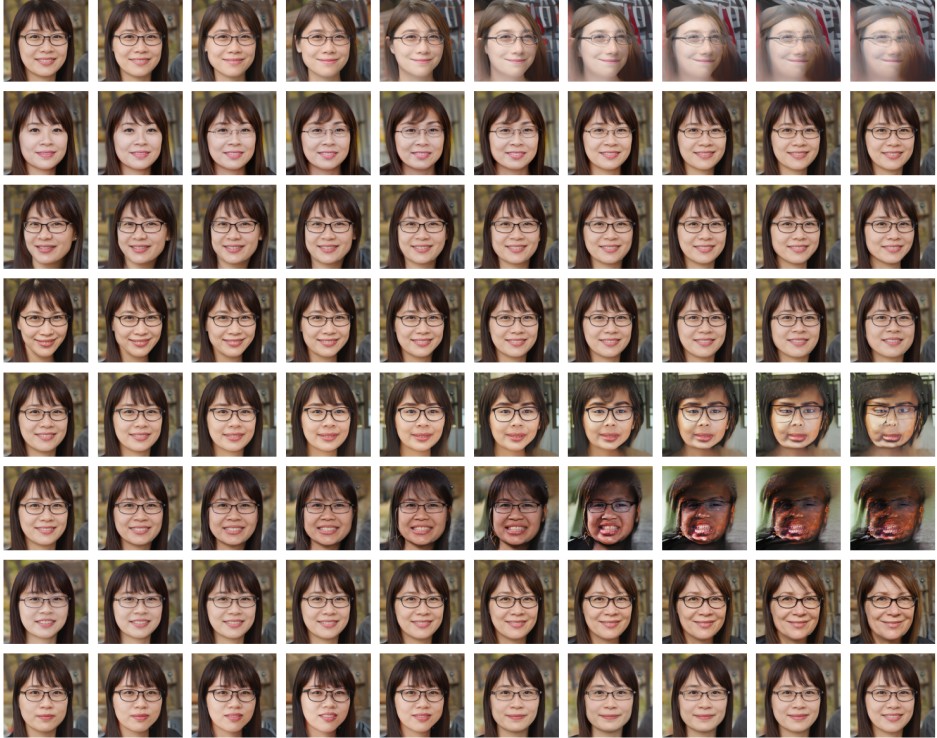

Figure 18: Traversals in the control space of PluGeN with $K$ controls: $(1)gender$, $(2)glasses$, $(3)yaw$, $(4)pitch$, $(5)baldness$, $(6)facial\ hair$, $(7)age$, $(8)expression$.

