# OpenReview forum: "DisCoVAE: Disentangling pretrained latent spaces with customized controls using contrastive learning"
_TMLR — Decision pending for TMLR_

### Review · Reviewer_DLdT · 2026-05-08

**Summary Of Contributions:**

This paper proposes DisCoVAE, a method for adding user-defined controls to pretrained generative models by learning a VAE-based control space over the pretrained latent space. The key idea is to use supervised contrastive learning on variation embeddings to disentangle different control factors. The paper provides two settings: learning multiple controls jointly and adding controls iteratively one by one. Experiments on DSprites, 3DShapes, and FFHQ show improved disentanglement and controllable generation compared with prior methods.

**Additional Comments:**

The paper is well written and the experiments are generally clear. I recommend acceptance, with revisions mainly focused on clarifying assumptions, limitations, and real-world robustness.

**Audience:**

Yes

**Audience Explanation:**

The problem is important because pretrained generators often have good image quality but limited controllability. A method that adds customized controls without retraining the original generator is useful. The iterative setting is also practical because users may want to add only one control at a time.

**Broader Impact Concerns:**

The method improves controllability of generative models, which can be useful for creative applications and data augmentation. However, face attribute editing may also be misused to manipulate identity-related or sensitive attributes. The paper should briefly discuss this risk.

**Claims And Evidence:**

Yes

**Claims Explanation:**

The comparisons with TwoStageVAE, DisCo, and PluGeN are relevant and generally convincing. The results show that the proposed contrastive regularization improves the pretrained latent space and enables better control. However, the evidence is weaker on real-world FFHQ attributes such as age, yaw, pitch, and baldness. The paper should more clearly discuss these limitations.

**Requested Changes:**

1. Clarify the scope of the method. The claim that it applies to any generative model with an exposed latent space seems too broad.

2. Clarify the supervision requirement. Method 2 does not need labels for all factors, but it still needs labels for the target factor.

3. Add or discuss ablations against simpler supervised losses, such as direct classification/regression or standard supervised contrastive learning.

4. Provide more analysis on FFHQ, especially for weak or imbalanced attributes. Balanced accuracy or macro-F1 would be helpful.

5. Discuss whether the iterative Method 2 is sensitive to the order in which controls are added.

6. Clarify the assumptions behind Theorem 1, especially because real datasets may have correlated or imbalanced attributes.

---

> ### Author Response · Authors · 2026-06-03
> **Detailed response**
>
> We thank the reviewer for the positive assessment of the paper and for recognizing both the practical value of the iterative setting and the ability of the proposed framework to add customized controls on top of pretrained generative models.
>
> **Scope of the method**
>
> We agree that the statement that DisCoVAE applies to "any generative model" is too broad. More precisely, the method requires access to an exposed latent representation together with an encoder or latent inversion mechanism that allows samples to be mapped into this latent space. We will revise the manuscript accordingly and clarify the assumptions under which the proposed framework can be applied.
>
> **Supervision requirements**
>
> We agree that this point should be clarified. While Method 2 does not require labels for all generative factors simultaneously, it still relies on supervision for the target attribute being disentangled. The intended contribution is to reduce the annotation burden by allowing controls to be learned incrementally, one feature at a time, rather than requiring complete multi-attribute annotations. We will make this distinction more explicit throughout the manuscript.
>
> **Comparison with simpler supervised objectives**
>
> We thank the reviewer for this suggestion. We note that the feature extraction experiments reported in Tables 2 and 3 already compare DisCoVAE against a supervised multi-head classifier trained directly on the target attributes. Regarding supervised contrastive learning (Khosla et al., 2020), we agree that this would constitute an interesting additional comparison. However, our method does not simply apply a supervised contrastive loss to the control embeddings themselves. The key contribution lies in the proposed variation-space formulation, where contrastive learning is performed on embedding differences constructed from groups of samples. This design is intended to explicitly model control-specific variations and promote disentanglement through the learned control distributions. We will clarify this distinction in the revised manuscript and discuss standard supervised contrastive objectives as a related alternative formulation.
>
> **Analysis of FFHQ and imbalanced attributes**
>
> The experiments already highlight that real-world attributes are affected by class imbalance, attribute correlations, and dataset biases which makes disentanglement substantially more challenging than on synthetic benchmarks.We partially discuss this in Section 5.2.3 and provide a more detailed analysis in Appendix A.3.2. (due to page limits).
>
> **Sensitivity of Method 2 to the order of controls**
>
> We agree that this aspect deserves further discussion. Our experiments indicate that the order in which controls are added can influence the resulting representation when attributes are correlated. This phenomenon is already visible in the analyses reported in Appendix A.3.2, where correlations such as age–glasses and facial-hair–gender affect the learned controls. While the iterative procedure remains effective overall, these observations highlight a limitation of sequential learning in the presence of strong dataset biases. We will make this limitation more explicit.
>
> **Assumptions behind Theorem 1**
>
> We agree that the assumptions behind Theorem 1 should be clarified. The theorem establishes an expectation result under the sampling procedure described in Section 3.2.2 and should not be interpreted as guaranteeing perfect disentanglement in real-world datasets. In practice, correlated attributes, class imbalance, and sampling biases may violate the idealized assumptions underlying the analysis. We will clarify these assumptions and better distinguish the theoretical intuition from the practical limitations observed in the experiments.
>
> **Broader impact**
>
> We thank the reviewer for raising this point. We agree that improved controllability of generative models may facilitate misuse, particularly in the context of face-attribute manipulation. We already included a broader impact section after the conclusion, where we discuss potential risks related to deceptive image editing, manipulation of sensitive attributes, and the amplification of societal biases present in the training data. We also discuss possible mitigation strategies, including auditing latent directions for bias, restricting manipulations involving sensitive attributes, and clearly communicating the limitations of learned latent controls.

---

### Review · Reviewer_G2fe · 2026-05-11

**Summary Of Contributions:**

The paper proposes a VAE-based module trained on top of pretrained generative model, producing an mapping between the original entangled latent space and a new disentangled control space. The disentanglement is driven by a contrastive (InfoNCE) objective which forces each labeled attribute onto its own latent dimension and turns discrete labels into continuous Gaussian-mixture controls. The method 2 (M2) variant disentangles one attribute at a time using only a single label, enabling an iterative workflow that incrementally builds up a personalized control space. The method is evaluated on datasets (DSprites, 3DShapes, FFHQ)

**Audience:**

Yes

**Audience Explanation:**

The answer to this question is twoflod: the topic analyzed by the authors is definetely an important one, giving a lightweight method to adapt the pretrained generative models and possess greater controllability over its output is still an open problem and I believe that many readers will find it interesting. That being said, I believe that the current evaluation is insufficient to actually assess the practicality of the method.

Three things I'd like to highlight:
1) The topic of controllability is dynamic field with multiple different ideas which are entirely skipped in both related works (which itself is too short) and in the evaluation.
2) Since the idea is model-agnostic it would be beneficial for the work to test it on different models except for the VAE pretrained on the synthetic datasets.
3) The whole work gives a feeling of being a bit outdated (the experiments, baselines, pretrained models) and thus make it hard to actually reasonably assess the relevance of the method. To be more precise I've noticed that two latest references in the whole work dates back to 2024 and 2023. A lot has changed since then (see for instance SAE approach or concept steering for reference).

**Claims And Evidence:**

No

**Claims Explanation:**

The work is generally clearly presented and easy to follow accompanied by an experimental section to support the results as well as theoretical justification for the method. However, the experimental part is what makes the work relatively weak. While the authors claim that " (...) method strongly outperforms all baselines (...)" the quality of these baselines leaves much room for improvement. As the authors themselves acknowledge one of the baselines "(...) TwoStageVAE can also be considered as an ablation study (...)" of the proposed method. Thus from my perspective it is hard to judge whether the proposed method offers anything new compared what's already established.

**Requested Changes:**

Properly frame the contribution within the field (reference up to date works).

Compare the results with recent publications. What about ohter variants of adaptability? PluGen is definetely insufficient to compare with.

Apply the method (and comparative baselines) to other architectures beyond the small datasets/models.

---

> ### Author Response · Authors · 2026-06-03
> **Detailed response**
>
> We thank the reviewer for the thoughtful feedback and for recognizing the importance of controllability in pretrained generative models.
>
> **Positioning within recent controllable generative modeling research**
>
> We agree that the manuscript would benefit from a broader discussion of recent developments in controllable generative modeling. Our goal is not to compete with methods that directly steer generation through guidance mechanisms, concept editing, sparse representations, or model-specific architectural modifications, but rather to address a complementary problem: learning a disentangled and user-customizable control space on top of an existing pretrained latent representation.
>
> In this sense, DisCoVAE can be viewed as a latent-space restructuring approach rather than a steering approach. While recent diffusion-based methods increasingly rely on guidance, concept steering, and latent editing techniques that exploit semantic directions already present in a pretrained representation, DisCoVAE explicitly reorganizes the latent space into independent user-defined controls. The distinction becomes particularly relevant when the pretrained latent manifold is highly entangled. As shown by the results on DSprites (Table 1), DisCo, which relies on local latent-space shifts to uncover disentangled directions and therefore implicitly assumes a degree of local linearity, struggles to recover meaningful factors from the pretrained latent space. In contrast, DisCoVAE successfully disentangles the targeted controls by explicitly restructuring the latent representation through the proposed contrastive objective.
>
> Importantly, the central contribution of the paper is not tied to a particular generation paradigm. Our FFHQ experiments already rely on DiffAE (Preechakul et al., 2022), a diffusion-based autoencoding framework, illustrating that the proposed control-space learning mechanism is not restricted to conventional VAEs and can be applied to latent representations derived from diffusion models. We will expand the related work section to better position DisCoVAE with respect to recent controllable diffusion, concept steering, latent editing, and representation-learning approaches.
>
> **Choice of baselines and relevance of the evaluation**
>
> We respectfully believe that PluGeN, DisCo, and TwoStageVAE represent the most relevant baselines for evaluating the specific contribution of DisCoVAE.
>
> PluGeN addresses the same problem setting of introducing controllable factors into a pretrained latent space through an explicit latent-space transformation. DisCo is particularly relevant because it also leverages contrastive learning to identify disentangled directions in pretrained latent spaces. In contrast, DisCoVAE uses contrastive learning not to discover existing directions, but to explicitly reshape the latent representation into a user-defined disentangled control space. Finally, TwoStageVAE serves as a direct ablation of our framework by removing the proposed contrastive regularization while preserving the same VAE-based latent restructuring architecture.
>
> These baselines therefore isolate the main design choices underlying DisCoVAE and allow us to assess the contribution of the proposed contrastive variation-space objective independently of architectural differences. In particular, while TwoStageVAE shares the idea of learning a secondary latent representation on top of a pretrained latent space, the key novelty of DisCoVAE lies in the proposed variation-space contrastive objective and the iterative disentanglement procedure. These additions fundamentally change the objective from density modeling of latent representations to the learning of user-defined disentangled controls. Empirically, the  performance gap between TwoStageVAE and DisCoVAE across disentanglement metrics and controllability tasks indicates that the gains cannot be attributed solely to the use of a second VAE. We will clarify this distinction more explicitly in the revised manuscript.
>
> **Evaluation on additional architectures**
>
> We agree that evaluating additional architectures would be valuable and constitutes an important direction for future work. However, we would like to emphasize that the current evaluation already includes both convolutional VAE latent spaces (DSprites and 3DShapes) and a diffusion-based latent representation through DiffAE on FFHQ. The purpose of these experiments was precisely to demonstrate that the proposed control-learning mechanism is not tied to a specific generative architecture but rather operates on pretrained latent representations. We will clarify this point in the revised manuscript and discuss broader evaluations on additional architectures as future work.

---

### Review · Reviewer_Fogw · 2026-05-12

**Summary Of Contributions:**

The paper introduces DisCoVAE -- a method for disentangling latent spaces of generative models. The main goal is to learn how to control the generative process of various models by editing or conditioning on specific values of certain images while keeping the rest unchanged, e.g., change the size of a square without changing its shape or color. The authors' proposed method relies on applying a VAE on the existing pre-trained latent space. On top of the new VAE latent space in turn they construct a variation space, which is then structured using contrastive learning objectives -- the authors propose two different approaches they call M1 and M2. The authors show how DisCoVAE performs on a suite of standard disentanglement datasets as well as FFHQ, a more realistic, high-dimensional dataset.


Overall, I think the paper is interesting and well-executed, making it a great fit for TMLR.

Strengths:
- Interesting approach to the disentanglement problem that combines latent space structuring with contrastive learning.
- The paper includes a wide array of experimental results, showing that the method performs quite strongly.
- The proposed method allows one to iteratively introduce additional control dimensions rather than training all of them at once, giving the user more freedom.
- The paper is well-written and the discussed related work is mostly sufficient.

Weaknesses:
- The paper doesn't discuss how the field of conditional generative models changed over the past few years. Currently, diffusion models are by far the most common models used in the industry as well as SOTA on many practical tasks, but they are barely discussed in the paper. I think it's worthwhile to focus on research that is not currently dominating the field, but I think the paper should do a better job of explaining the connections to the most popular methods.
- I am somewhat skeptical of how well the method scales. The results on dSprites and 3d-shapes are better than Plugen, but then they are not so conclusive on the relatively more complex FFHQ dataset (see e.g., Table 4). One can argue that FFHQ is actually the simpler dataset due to having mostly binary attributes -- but then it would be nice to get a better understanding of where DisCoVAE shines and where it underperforms.
- It seems the method doesn't have a way to change attributes in a directed way -- i.e., we can change a dimension we know should be changed (e.g., one related to age), but we do not know beforehand what's the direction (or range) we should explore. I'd suggest discussing this in the limitations section.

**Audience:**

Yes

**Audience Explanation:**

Conditional generative modeling is an active field of research so I'd expect there is a significant audience of TMLR readers who would find the paper interesting. However, I'll reiterate my comment that a discussion of current SOTA of generative models, including diffusion models, would make the paper stronger.

**Claims And Evidence:**

Yes

**Claims Explanation:**

As far as I can tell, all claims provided by the authors are well-supported. The authors do not overstate their contributions in the Introduction.

One small note:
- The final contribution in the Introduction says "We prove an invertible mapping..." -- it is not really invertible. With VAE one can get an approximation of the inverse, which is usually fine, but in this setting I'd read it as "perfectly invertible", i.e., the effect one can obtain with a normalizing flow.

**Requested Changes:**

I'm satisfied with the current form of the paper. However, there are a few points mentioned in the sections above that would strengthen it. Namely:
- Discuss SOTA generative models and how they relate to DisCoVAE.
- Provide further discussion about which settings work best for DisCoVAE and where it might underperform.
- Fix the claim about the function being invertible in the last contribution in the introduction section.

---

> ### Author Response · Authors · 2026-06-03
> **Detailed response**
>
> We thank the reviewer for the positive assessment and for recognizing the practical advantages of our proposed lightweight framework to iteratively add control on pretrained generative models.
>
> **Discussion of diffusion models and current generative modeling trends.**
>
> We agree that the connection between DisCoVAE and recent diffusion-based controllable generation methods should be discussed more explicitly. While our work focuses on learning a disentangled control space on top of a pretrained latent representation, recent diffusion-based approaches have increasingly relied on guidance, concept steering, and latent editing techniques to control generation after pretraining. These methods typically manipulate the generation trajectory or exploit semantic directions already present in the representation space to steer attributes of interest, whereas DisCoVAE aims to explicitly restructure the latent space into independent user-defined controls.
>
> The two directions are therefore complementary. Steering methods assume that meaningful semantic directions can be identified in the pretrained representation, while our approach learns a new control representation that isolates targeted factors and provides a bidirectional mapping between the original latent space and the learned control space. This distinction is particularly relevant when the pretrained latent manifold is highly entangled. As shown by the results on DSprites (Table 1), DisCo, which relies on local latent-space shifts to uncover disentangled directions and therefore implicitly assumes a degree of local linearity, struggles to recover meaningful factors from the pretrained latent space. In contrast, DisCoVAE successfully disentangles the targeted controls by explicitly restructuring the latent representation through the proposed contrastive objective.
>
> Our FFHQ experiments already rely on DiffAE (Preechakul et al., 2022), a diffusion-based autoencoding framework, illustrating that the proposed control-space learning mechanism is not restricted to VAEs and can be applied to latent representations derived from diffusion models. In the revision, we will expand the discussion of recent controllable diffusion methods, including guidance, concept steering, and latent editing approaches, and clarify how DisCoVAE relates to these developments. From this perspective, DisCoVAE can be viewed as a latent-space restructuring approach rather than a steering approach.
>
> **Where DisCoVAE works best and where it may underperform.**
>
> Our results suggest that DisCoVAE is particularly effective in settings where the pretrained latent space is highly entangled and the goal is to introduce user-defined controls without retraining the original generative model. Unlike direction-discovery or steering approaches that rely on identifying semantic directions already present in the latent space, DisCoVAE explicitly restructures the representation into a disentangled control space. This is illustrated by the results on DSprites (Table 1), where our method successfully disentangles the targeted factors despite the highly entangled nonlinear latent manifold, whereas DisCo struggles to recover meaningful directions under the same conditions.
>
> Our approach also appears more robust to heterogeneous control types and varying numbers of attribute classes. In particular, the feature extraction results on DSprites and 3DShapes (Tables 2 and 3) show that performance remains relatively stable as the number of classes increases, whereas PluGeN's performance degrades substantially for attributes with larger cardinality. Additionally, the iterative formulation (M2) enables users to introduce controls incrementally using only labels for the target attribute, making the framework practical in scenarios where annotations are incomplete or become available progressively. This flexibility is enabled by a lightweight VAE architecture trained on top of the pretrained latent representation, avoiding any retraining of the underlying generator.
>
> At the same time, our experiments reveal several limitations. First, the method is sensitive to dataset biases and correlations between attributes. As discussed in Appendix A.3.2, correlations such as age–glasses or facial-hair–gender can lead the model to associate multiple semantic factors, which may affect disentanglement quality. This phenomenon is also reflected in the iterative setting, where the order in which controls are learned can influence the resulting representation, as shown in the analyses reported in Tables 19 and 20. Second, some factors may not be naturally represented by a single latent dimension. For example, the orientation factor in DSprites remains challenging despite the proposed disentanglement objective, suggesting that enforcing a strictly one-dimensional representation may not always be optimal.

---

> > ### Author Response · Authors · 2026-06-03
> > **Detailed response**
> >
> > **Directed control and traversal direction**
> >
> > DisCoVAE disentangles factors into dedicated dimensions but does not explicitly enforce a semantic ordering of the learned control distributions during training. As suggested, we will explicitly add this as a limitation. However, after training, the parameters of the Gaussian mixture associated with each control dimension can be inferred from the labeled samples, providing a semantic interpretation of the different modes and enabling the identification of meaningful traversal directions. In practice, this allows users to navigate the control space by moving between mixture components and interpolating between their estimated parameters. Nevertheless, the method does not guarantee that a monotonic increase of a coordinate always corresponds to a monotonic increase of the underlying semantic attribute. We will clarify this point and discuss its implications for practical control in the revised manuscript.
> >
> > **Invertibility claim**
> >
> > We agree with the reviewer. The current wording may suggest an exact invertible mapping similar to normalizing-flow models. More precisely, our method learns a “bidirectional” mapping through the VAE encoder and decoder that approximately reconstructs the pretrained latent representation. We will revise the contribution statement accordingly to avoid overstating this property.

---

### Author Response · Authors · 2026-06-03
**Overall response to reviewers**

We thank all reviewers for their constructive feedback and thoughtful suggestions. We are encouraged that the reviewers find the problem of controllability in pretrained generative models important and recognize the practical value of a lightweight framework that can iteratively introduce user-defined controls without retraining the underlying generator.

The reviewers raised several useful points regarding the positioning of DisCoVAE within the broader controllable generative modeling literature, the scope and assumptions of the proposed framework, and the discussion of limitations on real-world datasets. We agree that these aspects can be clarified and strengthened. In the revised manuscript, we will:

- expand the discussion of recent controllable generation approaches, including diffusion-based guidance, concept steering, and latent editing methods
- better position DisCoVAE as a latent-space restructuring approach that complements existing steering-based methods
- clarify the assumptions underlying Method 2 and Theorem 1, as well as the supervision requirements of the iterative framework
- refine the discussion of strengths and limitations, including the effects of attribute correlations, dataset biases, and control-order sensitivity
- revise several statements that may currently be too broad, including the scope of applicability of the method and replace “invertible” by “bidirectional mapping”

We would also like to emphasize that the main contribution of DisCoVAE is not tied to a specific generative architecture, but rather to the problem of learning a disentangled and customizable control space on top of pretrained latent representations. This is reflected in our evaluation, which includes both VAE-based latent spaces and a diffusion-based latent representation through DiffAE. We appreciate the reviewers' suggestions and believe that the proposed revisions will further improve the clarity and positioning of the paper.